# Modeling cyanobacteria life cycle dynamics and historical nitrogen fixation in the Baltic proper

Jenny Hieronymus[1], Kari Eilola[1], Malin Olofsson[1,2], Inga Hense[3], H. E. Markus Meier[4], Elin Almroth-Rosell[1]

[1]Department of research and development, Swedish Meteorological and Hydrological Institute, 60175 Norrköping, Sweden
[2]Department of Aquatic Sciences and Assessment, Swedish University of Agricultural Sciences, 750 07 Uppsala, Sweden
[3]Institute of Marine Ecosystem and Fishery Science, Universität Hamburg, 22767 Hamburg, Germany.
[4]Department of Physical Oceanography and Instrumentation, Leibniz Institute for Baltic Sea Research Warnemünde, 18119 Rostock, Germany

*Correspondence to*: Jenny Hieronymus (jenny.hieronymus@smhi.se)

**Abstract.** Dense blooms of filamentous diazotrophic cyanobacteria are formed every summer in the Baltic Sea. These autotrophic organisms may bypass nitrogen limitation by performing nitrogen fixation, which also governs surrounding organisms by increasing bioavailable nitrogen. The magnitude of the nitrogen fixation is important to estimate from a management perspective since this might counteract eutrophication reduction measures. Here, a cyanobacteria life cycle model has been implemented for the first time in a high-resolution 3D coupled physical and biogeochemical model of the Baltic Sea, spanning the years 1850-2008. The explicit consideration of life cycle dynamics and transitions significantly improves the representation of the cyanobacterial phenological patterns compared to earlier 3D-modelling efforts. Now, the rapid increase and decrease of cyanobacteria in the Baltic Sea is well captured, and the seasonal timing is in concert with observations. The current improvement also had a large effect on the nitrogen fixation load and is now in agreement with estimates based on *in situ* measurements. By performing four phosphorus sensitivity runs, we demonstrate the importance of both organic and inorganic phosphorus availability for historical cyanobacterial biomass estimates. The model combination can be used to continuously predict internal nitrogen loads via nitrogen fixation in Baltic Sea ecosystem management, which is of extra importance in a future ocean with changed conditions for the filamentous cyanobacteria.

## 1 Introduction

Bioavailable nitrogen is globally limiting primary production in the ocean (Moore et al., 2013). Diazotrophic cyanobacteria can bypass this limitation by performing nitrogen fixation. In addition, they may release up to 50% of their newly fixed nitrogen, which stimulates surrounding organisms (Wannicke et al., 2009; Ploug et al., 2010; 2011). Anthropogenic pressures and climate change synergistically affect the Earth's ecosystems (Steffen et al., 2015). As nitrogen-fixing cyanobacteria are suggested to be enhanced by elevated temperatures (Paerl and Huisman, 2008; Wannicke et al., 2018), there is an increasing need to further understand their bloom dynamics and ecosystem impact.

The Baltic Sea is a semi-enclosed brackish water body exposed to significant impacts from eutrophication (HELCOM, 2010) because of the combination of a large increase of nutrient supplies since World War II (Gustafsson et al., 2012), permanent stratification (e.g., Leppäranta, 2009), and long water residence times (Meier, 2007), which reduce the deep water ventilation and enhance the widespread oxygen deficiency. Therefore, nations around the Baltic Sea have decided on a Baltic Sea Action Plan to reduce external loads of nutrients to the area (HELCOM 2007). The early history of multi-stressors and long-term data series in the Baltic Sea provides an opportunity to study consequences and possible mitigation strategies for future management of aquatic systems (Reusch et al. 2018).

Dense blooms of diazotrophic filamentous cyanobacteria, dominated by the taxa *Nodularia spumigena*, *Aphanizomenon* sp., and *Dolichospermum* spp. (Klawonn et al., 2016), are formed every summer in the Baltic Sea (Kahru and Elmgren, 2014; Olofsson et al., 2020; 2021). Despite reduced nutrient inputs (Gustafsson et al., 2012), there has been an increase in their abundance during recent decades (Finni et al., 2001; Kahru and Elmgren, 2014; Reusch et al., 2018) which in turn, contributes to the eutrophication problem. Furthermore, future scenarios predict an earlier initiation of the phytoplankton spring bloom, and thus, a potentially prolonged growth period for filamentous cyanobacteria (Sommer et al., 2012; Kahru et

al., 2016) and results from four decades of monitoring in the Baltic Sea suggest basin-specific changes of the cyanobacteria abundance and species composition due to decreased salinities and elevated temperatures (Olofsson et al., 2020).

The processes involved in the bloom formation of filamentous cyanobacteria are not yet fully understood (e.g., Conley, 2009; Nausch, 2012; Wasmund, 2017) but recent model studies (Hense and Beckmann, 2010; Hense and Burchard, 2010) and observations (Suikkanen et al., 2010) indicate that the life cycle of cyanobacteria plays an important role in determining the timing, duration, and magnitude of the blooms. The Cyanobacteria Life Cycle (CLC) model was introduced by Hense and Beckmann (2006) and includes, in its original design, four life cycle stages representing a vegetative non-nitrogen-fixing

stage, a vegetative nitrogen-fixing stage, a resting stage (akinetes), and a non-growing recruiting stage. The rapid increase (or decrease) of the summer concentrations is, in the CLC model, a result of transfer between life cycle stages, in turn, dependent on light, temperature, and dissolved inorganic nitrogen (DIN; ammonium and nitrate). Phosphorus limitation is, however, not considered in the model formulation and still needs to be determined.

Since introduced, the CLC model has been further developed and studied (e.g., Hense and Beckmann, 2010; Hense and Burchard, 2010; Hense et al., 2013). In Hense and Burchard (2010) and Hense et al. (2013), the CLC model is implemented in a one dimensional water column model representing the Eastern Gotland basin, and the results show a clear improvement in the timing and duration of blooms compared to conventional one compartment models. While the aforementioned studies compared the model results to the seasonal cycle of cyanobacteria biomass, no comparison was made to observations of

nitrogen fixation. Furthermore, the effect of the CLC model on the nutrient composition of ambient water was not made, and it has not yet been tested for the entire Baltic Proper where blooms of filamentous cyanobacteria are dominating the summer phytoplankton blooms.

While phosphorus limitation has not previously been considered in the CLC model, studies show that the growth of filamentous cyanobacteria is sensitive to the availability of phosphate (Moisander et al., 2007; Olofsson et al., 2016) with some cyanobacterial taxa being able to utilize both phosphate and organic phosphorus (Schoffelen et al., 2018). Understanding the dependence of phosphorus by cyanobacteria is of great importance in order to adequately model the phosphorus dynamics in the water column and the effect on other phytoplankton functional types. Furthermore, the cyanobacterial phosphorus dependence has implications for management, as changes in phosphorus loads ultimately affect the input of nitrogen through nitrogen fixation. Phosphorus loads are therefore of extra importance to decrease to allowable levels suggested by the Baltic Marine Environment Protection Commission HELCOM (2018).

The aims of the current study were to gain an understanding in phosphorus dynamics in the Baltic proper as well as demonstrate the workings and boundaries of the CLC model in order to use it for continuous monitoring and estimates of nitrogen fixation for management purposes. This was done by: I) running sensitivity experiments addressing phosphorus limitation to determine the optimum settings for the Baltic proper in relation to cyanobacteria blooms; II) including the CLC model in a high-resolution 3D-coupled physical and biogeochemical model of the Baltic Sea; and III) comparing the new CLC model setup to the original model that excludes the CLC, and validating it to observations of cyanobacteria carbon biomass and estimated nitrogen fixation measurements based on previous *in situ* measurements.

**2 Method**

The Baltic Sea is a semi-enclosed estuary that has limited water exchange with the adjacent North Sea (Fig. 1). In order to study bloom formations of filamentous cyanobacteria, we included a modified version of the cyanobacteria life cycle (CLC) model in a high-resolution three-dimensional (3D) coupled physical-biogeochemical model of the Baltic Sea (Meier et al., 2003; Eilola et al., 2009; Almroth-Rosell et al., 2011) spanning 1850-2008. The CLC model is described in detail below, together with modifications of the biogeochemical model setup (schematically shown in Fig. 2, Table 1).

**2.1 Ocean circulation model**

The RCO (Rossby Centre Ocean) model is a Bryan–Cox–Semtner primitive equation circulation model with a free surface (Killworth et al., 1991). Its open boundary conditions are implemented in northern Kattegat, based on prescribed sea level elevation at the lateral boundary (Stevens, 1990). An Orlanski radiation condition (Orlanski, 1976) is used to address the case of outflow, and the temperature and salinity variables are nudged toward climatologically annual mean profiles to deal with inflows (Meier, 2003). A Hibler-type dynamic–thermodynamic sea ice model (Hibler, 1979) with elastic–viscous–plastic rheology (Hunke and Dukowicz, 1997) and a two-equation turbulence closure scheme of the k-$\varepsilon$ type with flux boundary conditions (Meier et al., 2001) is embedded into RCO. The deep-water mixing is assumed inversely proportional to the Brunt–Väisälä frequency, with the proportionality factor based on dissipation measurements in the Eastern Gotland Basin (Lass et al., 2003). RCO was used with a horizontal resolution of 2 nautical miles (3.7 km) and 83 vertical levels, with a layer thickness of 3 m. RCO allows direct communication between bottom boxes of the step-like topography (Beckmann and Döscher, 1997). A flux-corrected, monotonicity-preserving transport (FCT) scheme is applied in RCO (Gerdes, 1991). RCO has no explicit horizontal diffusion. For further details of the model setup, the reader is referred to Meier (2003; 2007).

The model performance of temperature and salinity was evaluated in Meier et al. (2018) and conformed well to observations but showed a higher position of the halocline and slightly lower bottom water salinity, and some deviations with higher temperatures were found in the upper part of the halocline.

**2.2 Biogeochemical model**

The biogeochemical model SCOBI (Swedish Coastal and Ocean Biogeochemical model) has been developed to study the nutrient cycling in the Baltic Sea (Marmefelt et al., 1999; Eilola et al., 2009; Almroth-Rosell et al., 2011; 2015). SCOBI handles biological and ecological processes in the sea as well as sediment nutrient dynamics and is in this study coupled to RCO (e.g., Eilola et al., 2012; 2013; 2014). Resuspension of organic matter is calculated, with the help of a simplified wave

model, from the wave and current-induced shear stresses (Almroth-Rosell et al., 2011). The model includes three different

functional types representing diatoms, flagellates and other microalgae, and cyanobacteria as well as one zooplankton. SCOBI has a constant carbon to chlorophyll a ratio, 50 mg C (mg Chl)$^{-1}$, and the production of phytoplankton assimilates carbon, nitrogen, and phosphorus according to the Redfield molar ratio (106:16:1, respectively) (Eilola et al., 2009). The molar ratio of complete oxidation of the remineralized nutrients is $O_2$:carbon = 138. Nitrogen fixation is a function of temperature, light availability, nitrogen to phosphorus ratio, and phosphorus concentration in the ambient water (Eilola et al.,

2009). Dead organic material, represented by separate variables for nitrogen and phosphorus accumulates in detritus in the water column and in the sediments. For further details of the "standard" SCOBI model, the reader is referred to Eilola et al. (2009; 2011) and Almroth-Rosell et al. (2011).

### 2.3 Cyanobacteria life cycle model

In the original SCOBI model, cyanobacteria are represented by one state variable and the population is upheld by a minimum

biomass. The growth is dependent on light, temperature, and nutrients (nitrogen and phosphorus), and nitrogen fixation occurs when the DIN concentration isare low after the spring bloom.

The CLC model is seasonal with only one full life cycle each year (Fig. 3) and simulates four state variables instead of one in the original SCOBI model. It is modified from the detailed life cycle model by Hense and Beckmann (2006) that includes

internal nitrogen and energy quotas, and the simplified version by Hense and Beckmann (2010). Similar to Hense and Beckmann (2010), growth and life cycle transitions in our CLC model depend only on external factors, but we kept the sinking and rising stage separated, generating an additional life cycle variable. The CLC model equations as well as variables and parameters can be found in table S.3 in the Supplementary material.

We distinguish between three life cycle stages ( Fig. 3): the growing and nitrogen-fixing stage (vegetative cells with heterocysts, HET), the resting stage (akinetes, AKI) and a stage (REC) where we combine the recruiting (cells with gas

vesicles) and the growing, non-nitrogen-fixing stage (vegetative cells without heterocysts). REC and HET are assumed to aquire carbon, nitrogen and phosphorus, and produce detritus, according to Redfield molar ratios. HETs are positively buoyant, AKIs in the water (AKIW) are sinking and may end up in the sediment (AKIB) and RECs are rising. Dead HETs and RECs end up in the pool of dead organic matter (Fig. 2). Occasions with resuspension may transfer akinetes from the sediment (AKIB) to the water (AKIW).

Life cycle transitions were treated in a relatively simple way: Following Hense and Beckmann (2010), we used the *in situ* growth rate for the transition between the life cycle stages HET and AKI. In autumn, when the growth rate, that is dependent on temperature and light, was below a critical threshold, a transfer into the AKI compartment took place (Eq. (11), Table S.3).

For the transition between AKI (AKIB and AKIW) and REC we prescribed a fixed germination window instead of using a dynamic germination window as proposed by Hense and Beckmann (2010). Between April 20 and the end of April, germination occurred at a constant rate times the AKI concentration (cf. Eq. (30) in Table S.3). This is because the computational costs of a dynamic window in a 3D framework are too high. However, shifting the germination window has only a small impact on the timing of maximum cyanobacteria abundance in summer and the magnitude of nitrogen fixation. A sensitivity test showed that the decadal mean nitrogen fixation was lower when germination was earlier by about 4-7%, while the maximum annual difference found for the entire period (1850-2008) was 14% lower.

The transition from the recruiting and vegetative state (REC) to the diazotrophic state (HET) takes place when the growth of HET is larger than that of REC (Eq (10) in Table S.3). The maximum growth rate ($s^{-1}$) of REC is larger than that of HET, but the growth (mmol $m^{-3}$ $s^{-1}$) is in the previous state, also dependent on DIN. When DIN is low after the spring bloom the growth of HET becomes larger than that of REC and a transition to HET occurs.

The temperature dependence of HET and REC growth is given by the temperature limitation function (Supplementary material, Table S.3, Eqs. 8 and 26). Between approximately 6°C and 28°C an increase in temperature positively affects the growth rate of HET and REC while an increase above 28°C instead generates a decline in growth rate. The model equation was designed to represent the observational data presented in Lethimäki et al. (1997). The original formulation in Hense and

Beckmann (2006), based on the same data, was found to be numerically unstable in the 3d model framework and was therefore redesigned. The resulting model formulation closely resembles that of Hense and Beckmann (2006) but gives slightly higher growth rates at low temperatures.

For potential growth of REC and HET and potential transition of AKI to REC, we assumed a salinity span between 3 and 10

PSU, which is in fair agreement with the optimum growth of *N. spumigena* (5-10 PSU), *Aphanizomenon* sp. (0-7 PSU) (Rakko and Seppälä, 2014), and *Dolichospermum* spp. (0-6 PSU) (Teikari et al., 2018). This constrains the cyanobacteria to areas within the Baltic Sea that lies within the given salinity span. The AKIB is assumed to be rapidly immobilized in the sediment, simulated as a very large burial, under salinity outside of the defined range.

Similar to Hense and Beckmann (2006; 2010), we pooled the three main important nitrogen-fixing taxa *N. spumigena*, *Aphanizomenon* sp. and *Dolichospermum* spp. into one functional cyanobacteria group. We are well aware that there are differences among the species (e.g., with respect to salinity and/or temperature dependence) and thus we may not expect to be able to reproduce specific local patterns, for example in a low-salinity region outside of the range where some of the taxa can still thrive. Nevertheless, as we will show, our model was able to reproduce the main seasonal and spatial patterns of

biomass and nitrogen fixation.

We herein refer to the 3D coupled RCO-SCOBI that includes the new CLC as SCOBI-CLC while the old model version without CLC will be referred to as SCOBI.

**2.4 Model forcing**

The historical simulation uses reconstructed atmospheric, hydrological, and nutrient load forcing, and daily sea levels at the lateral boundary for the period 1850-2008 as described in detail in Meier et al. (2018) and Gustafsson et al. (2012) and references therein. The used High Resolution Atmospheric Forcing Fields for the period 1850-2008 were reconstructed using atmospheric model data for 1958–2007 together with historical station data of daily sea-level pressure and monthly air temperature observations. For the calculation of monthly mean river flows five different historical data sets were merged. The basin integrated reconstructed nutrient loads from land and atmosphere to the present model are the same as used and described by Gustafsson et al. (2012). Nutrient loads contain both organic and inorganic phosphorus and nitrogen, respectively. In the present SCOBI version, the nitrogen and phosphorus detritus were separated and thus used both organic phosphorus and nitrogen from the forcing. This is the only difference in forcing from the present SCOBI model compared to the model used by Meier et al. (2018), where detritus consisted of one pool limited by the Redfield ratio. Daily mean sea level elevations at the boundary in the Northern Kattegat were calculated from the reconstructed, meridional sea level pressure gradient across the North Sea. In case of inflow, temperature, salinity, nutrients and detritus values were nudged towards observed climatological seasonal mean profiles for 1980–2005 at the monitoring station Å17 in the southern Skagerrak. Nutrient concentrations before 1900 were assumed to be only 85% of present-day concentrations. A linear decrease of nutrient concentrations from 1950 and back in time to 1900 was assumed.

**2.5 Observations**

The Swedish National Marine Monitoring Program includes monthly tube sampling of phytoplankton abundance (including filamentous cyanobacteria) and water collection for chemical and physical parameters (e.g., inorganic nutrients, oxygen, salinity, temperature). This data is hosted by the Swedish National Oceanographic Data Centre at the Swedish Hydrological and Meteorological Institute and is freely accessible at www.smhi.se. For this work, we also used data of oxygen and nutrients from The Baltic Environmental Database (BED), which includes post-processed monitoring station data from a number of institutes around the Baltic Sea. The data is freely available at http://nest.su.se/bed. The cyanobacteria biovolume

(mm$^3$ l$^{-1}$) was calculated based on cell numbers and size of filaments (Olenina et al., 2006) and further to carbon concentrations (referred to as cyanobacteria biomass) based on Menden-Deuer and Lessard (2000). Concentrations of inorganic nutrients and oxygen were extracted from the database for station BY15 in the Eastern Gotland Basin, and cyanobacteria biomass for four stations in the Baltic proper for 1999-2008 (Fig. 1).

The cyanobacteria biovolume was used to estimate nitrogen fixation rates (mmol N m$^{-2}$ d$^{-1}$) based on empirical biovolume-specific measurements (Klawonn et al., 2016) according to calculations in Olofsson et al. (2021). Shortly, the observed biovolumes (mm$^3$ l$^{-1}$) were multiplied with biovolume-specific measurements (µmol N mm$^{-3}$ d$^{-1}$), and further integrated over 0-10 m to obtain area-specific nitrogen fixation rates (mmol N m$^{-2}$ d$^{-1}$). These rates were summarized, first monthly and then for the whole year, and multiplied with the size of the Baltic Proper (200,000 km$^2$) to provide annual nitrogen loads via nitrogen fixation by filamentous cyanobacteria (kton N yr$^{-1}$) during 1999-2008.

## 2.6 Phosphorus dependence

In the original model by Hense and Beckmann (2006) that includes the internal energy and nitrogen, the seasonal changes in cyanobacteria biomass are adequately modelled without taking phosphate into account. The rapid decrease of HET in autumn is then a result of an internal energy crisis caused by the high energy demand of nitrogen fixation together with decreasing temperatures and light. This is also true for the simplified model of Hense and Beckmann (2010) where the growth rate of HET is strongly limited by temperature. However, in the Baltic Sea, the phosphorus concentrations may limit the cyanobacteria biomass (Klawonn et al., 2016; Degerholm et al., 2006; Olofsson et al. 2016). We have therefore performed four sensitivity runs, listed below, to evaluate the role of phosphorus uptake. We distinguish between uptake of inorganic and organic phosphorus, since both types are utilized by cyanobacteria (Schoffelen et al., 2018). The preferential uptake of dissolved inorganic phosphorus is, however, assumed in the model.

- **noP** - Phosphorus is excluded from cyanobacteria in line with Hense and Beckmann (2010). In this case, the cyanobacteria can grow completely independent of phosphate availability in ambient water. The cyanobacteria also do not, *in this case only*, take up or release any phosphate.

- **sPlim** - Strong limitations from both phosphate and organic phosphorus. In this case, the half saturation constants are large and the cyanobacteria growth depends strongly on the availability of phosphate.

- **wPlim** - A very small value ($10^{-6}$ mmol P m$^{-3}$) assigned to the half saturation constants of both the phosphate and the organic phosphorus limitation terms, effectively removing the phosphorus limitation of cyanobacteria. As long

as phosphate exists in small amounts in ambient water, the growth is maintained independent of the concentration. However, in the absence of phosphate the growth is terminated.

- **noOP** - Limitation by inorganic phosphate included but the ability to utilize organic phosphorus in cyanobacteria removed. As can be deduced from Eq. 1 in Supplementary material, Table. S.3, the growth, in this case, gets no

additional reinforcement from organic phosphorus.

The differences in parameter values between the phosphorus sensitivity runs are found in Table S.5 in the Supplementary material.

 **3 Results and discussion**

We start by presenting the results of cyanobacteria biomass and nutrients from the four different phosphorus limitation experiments using SCOBI-CLC. The optimum combination was then used to compare with our old model setup that does not include the CLC (SCOBI), and with *in situ* observations in Section 3.2.

**3.1 Phosphorus limitation experiments using SCOBI-CLC**

**3.1.1 Cyanobacteria biomass**

The simulated biomass was generally larger than observations in all four SCOBI-CLC phosphorus limitation experiments (Fig. 4). The experiment that generated the largest biomass was by far noP, which completely excludes the impact of phosphorus in ambient water, and where no uptake or release of phosphorus occurs. Since the cyanobacteria, in this case, are not dependent on phosphate, they grew extensively even in the first part of the 20th century (Fig. 5) when observations

indicate that cyanobacteria blooms seldom occurred (Finni et al., 2001). Through nitrogen fixation, the cyanobacteria also stimulate surrounding phytoplankton, generating higher biomass of these as well (Fig. 5, lower panel). Fig. 5 further displays a decline in cyanobacteria biomass from the mid 20th century to the 1980s for experiment noP in the central Baltic proper (BY15). The reason for this decline is most likely the sharp increase in nutrient loads (Gustafsson et al., 2012), generating a competitive advantage of faster-growing diatoms and flagellates leaving less DIN for the non-nitrogen-fixing RECs. This is

indicated also by the increase in diatoms and other phytoplankton biomass accompanying the cyanobacteria decline (Fig. 5, lower panel).

Up until about 1980, the noOP (no additional contribution from organic phosphorus) generally generated the lowest annual mean biomass and wPlim the highest after noP (Fig. 5). This is to be expected as the growth rate, given the same amount of

organic and inorganic phosphorus, in the wPlim case, is largest (see Eq. 1 and Eq. 4 in Table S.3). The experiments sPlim, wPlim, and noOP generated results closer to observations as compared to noP for the period 1999-2008 (Fig. 4). The lowest biomass was, for this period, generated through wPlim for all stations except BY31, despite generating the highest biomass

in the central Baltic proper up until 1980 (Fig. 5). It is notable that wPlim also generated the best bloom timing as compared to observations. This experiment allows cyanobacteria to grow quickly even at low phosphate concentrations as long as there is enough light, and the temperature is above approximately $6°C$ (see Eq. 8 in Table S.3). Thereby, light and temperature set the threshold of when the bloom will be initiated, and the phosphate dependence decides the end of it. The noP run generated a start of bloom that is close to wPlim but a later termination.

The observations show an increase in filamentous cyanobacteria biomass in May-June and with a maximum abundance in July-August (Fig. 4), which is a typical seasonal cycle of cyanobacteria in the Baltic proper (Olofsson et al., 2021). The experiment that showed a seasonal timing that best corresponds to observations is wPlim displaying a slightly earlier summer maximum than the other sensitivity experiments.

### 3.1.2 Nutrients and oxygen

Diazotrophic cyanobacteria increase bioavailable nitrogen in the water through their release of ammonium from its newly fixed nitrogen (Ploug, 2010; 2011). They also impact surrounding organisms by competing for phosphate. The influence of the cyanobacteria on the nutrient concentrations can therefore be seen in the different sensitivity experiments.

In order to understand the difference in biomass in the earlier and later part of the model run between the experiments noOP, wPlim and sPlim, we demonstrate the simulated mean seasonal cycle of phosphate and nitrate for the periods 1999-2008 and 1960-1980 for the central Baltic proper, BY15 (Fig. 6). In the earlier period, wPlim consistently generated the highest biomass, apart from noP, and noOP the lowest as expected from the lower growth rate obtained in this case. However, in the later part of the run, the biomass was lowest in wPlim.

In the early period, the mean seasonal cycles of nutrients in the central Baltic proper, BY15 (Fig. 6), show that the phosphate concentrations were higher and DIN concentrations lower in wPlim compared to the sPlim and noOP which generated higher cyanobacteria biomass. However, in the later period, the phosphate concentrations were lowest in wPlim generating a smaller biomass compared to the sPlim and noOP. Furthermore, during the later period (1999-2008), DIN was completely depleted after the spring bloom providing little opportunity for other phytoplankton than cyanobacteria to grow. During the earlier part of the run, DIN was available even during summer allowing for higher biomass of the surrounding diatoms and other phytoplankton (Fig. 4).

The surface winter concentration of phosphate and DIN as well as oxygen at 200 m depth at monitoring station BY15 were compared for the different runs together and observation data (Fig. 7). All experiments, with the notable exception of noP, conformed well to observed winter surface phosphate. With no phosphorus in cyanobacteria, the winter phosphate concentration becomes too high reflecting the extensive primary production that leads to deep water oxygen depletion and generates sedimentary phosphate release in this experiment. The 200 m oxygen concentration was well captured in all other experiments.

### 3.2 Nitrogen fixation and cyanobacteria biomass in SCOBI-CLC, SCOBI and observations

To estimate cyanobacteria biomass and nitrogen fixation rates and compare with observations for the Baltic proper, we used the phosphorus limitation setting wPlim based on the limitation experiments that best captured the size of biomass and the seasonal timing (Fig. 4). Compared to the SCOBI, SCOBI-CLC displays significant improvements in seasonal timing (Fig. 8). In line with observations, SCOBI-CLC generates a peak biomass in July, while the SCOBI results reach peak biomass in September. This is an important improvement attained by using the CLC model as compared to previous results (Hieronymus et al., 2018) as the seasonal timing of biomass also affects the timing and size of nitrogen fixation. By obtaining a bloom more constrained to the summer months, a larger nitrogen fixation due to higher temperatures was

observed using SCOBI-CLC. The updated nitrogen fixation rates were also in the same range as estimates based on measurements for the same stations during the years 1999-2008, both in magnitude and timing (Fig. 9).

For nitrogen fixation, there was a slight difference where SCOBI-CLC displayed a prolonged peak period in July - August while the observations showed a peak more constrained to July. The strong coherence between modelled and observed nitrogen fixation is somewhat surprising given the larger cyanobacteria biomass in SCOBI-CLC compared to observations (Fig. 8). There may be several reasons for this discrepancy. The frequency of observations is bimonthly at most, which occasionally means missed peak values. Furthermore, the cyanobacteria biovolume from observations was used with

different presumptions to estimate the nitrogen fixation rates and to calculate carbon concentrations, respectively. The modelled nitrogen fixation is calculated during the run from the growth of HET in the CLC model while the carbon content of cyanobacteria is calculated by the Redfield ratio between nitrogen and carbon in HET with a minor contribution also from REC. Hence, there are uncertainties in the calculations of carbon biomass from both observations and from model results. It is not easy to change the Redfield C:N:P ratio that is used in the model since the results from the entire biogeochemical cycle

including the oxygen consumption in the model is dependent on this ratio. There are other biogeochemical models with variable C:N:P ratios that might be used to analyze the impact from these processes further (e.g., Fransner et al., 2018; Kwiatkowski et al., 2018). Uncertainties in the comparison of models and observations stem also from the fact that observations are done on small water samples from an area that is covered by an average value from a 3.7 km x 3.7 km grid in the model.


Nitrogen fixation in the Baltic Sea is dominated by the three filamentous taxa described herein (Klawonn et al., 2016). However, heterotrophic nitrogen fixation has also been observed in the Baltic Sea (e.g., Farnelid et al., 2013), but since their rates are extremely low in this region, they do not affect the overall input of nitrogen (heterotrophic bacteria: up to 0.44 nmol $l^{-1}$ $d^{-1}$ in Farnelid et al., 2013, as compared to 800 nmol $l^{-1}$ $d^{-1}$ by the filamentous cyanobacteria in Klawonn et al., 2016).

We estimated the internal nitrogen load via nitrogen fixation to the Baltic proper based on monitoring and *in situ* measurements to a mean of 399 kton per year for 1999-2008, but with a large variation among years (SD ± 104). This is slightly below the external load from river runoff and atmospheric deposition of 430 kton per year (± 54), provided by HELCOM (2018). For SCOBI-CLC, we got an estimated mean nitrogen load of 409 kton per year for experiment wPlim over the same years for the Baltic proper (calculated over an area of 216,600 km$^2$) compared to 271 kton per year for SCOBI.

The estimated annual nitrogen load via nitrogen fixation to the Baltic proper has not changed over recent years (2013-2017 in Olofsson et al., 2021), and is in the range of other studies for the Baltic proper (310 kton in Rolff et al., 2007; 370 kton in Wasmund et al., 2001; 396 kton in Svedén et al., 2016) but below the estimated load of 613 kton in Wasmund et al. (2005), 514 kton in Gustafsson et al. (2013), and 511 kton in Schneider et al. (2009).

The mean vertical profiles of phosphate, DIN and oxygen at stations BY5 and BY15 showed an overall good representation by both SCOBI-CLC and SCOBI (Fig. 10). Below the mixed layer, the DIN concentrations were high compared to observations for SCOBI-CLC and the deep water phosphate at BY5, a bit too low. Surface phosphate is slightly closer to observations for SCOBI-CLC than for SCOBI. The low surface DIN in both model versions is a reflection of low nitrate concentrations as compared to observations which were also reported by Meier et al. (2012) and Saraiva et al. (2018). The 355 low surface DIN is also demonstrated in Fig. 7 where the noP experiment gave rise to higher DIN concentrations as nitrogen fixation due to strong cyanobacteria blooms, in this case, adds more DIN to the water column. Despite the shortcomings, the trends for both nutrients and oxygen were well captured by both SCOBI-CLC and SCOBI.

**4 Summary and conclusions**

Through a series of sensitivity experiments, we have shown that the inclusion of phosphorus dependence in cyanobacteria is essential for the CLC model in the Baltic proper but only a weak limitation is necessary. Excluding phosphorus in cyanobacteria generates too high biomass values, especially in the first part of the 20th century when cyanobacteria blooms

were rarely observed (Finni et al., 2001). The large primary production in this case was also reflected in too high phosphate concentrations as eutrophication induced anoxia which gave rise to sedimentary phosphate release.


By including the CLC model into a 3D model for the Baltic proper, we demonstrate a clear improvement in the seasonality of cyanobacteria blooms as compared to the old model which generates a peak biomass two months later and a nitrogen fixation peak one month later than observations. The next step in the development of the CLC model would be to include three individual types of cyanobacteria, to be able to more closely capture the differences between the dominating taxa (Klawonn et al., 2016). *Aphanizomenon* sp. for example can perform high nitrogen fixation rates already at 10°C during early summer (Svedén et al., 2015), and is responsible for the highest total nitrogen fixation in the region due to its long growth season (Klawonn et al., 2016). *Aphanizomenon* sp. may also use different sources of phosphorus, which may further separate the growth niches by the filamentous cyanobacterial species (Schoffelen et al., 2018). Phosphorus cycling is a complex topic, which also needs further studies in natural ecosystems, as high turnover rates of phosphorus of only about 2 h are hard to trace (Nausch et al., 2018). To include more species in the model might be of extra importance as climate change scenarios can change the community composition in the future (Wulff et al., 2018; Olofsson et al., 2020).

In this work, we have used a CLC model that includes benthic and pelagic akinetes from which the summer blooms originate. Research has shown that the life cycles of the different major bloom-forming taxa are complex and there is no single answer on how they start growing after winter (Munkes et al., 2021). Experiments have suggested that all taxa form akinetes to some extent but the summer bloom of *N. spumigena* and *Aphanizomenon* sp. originates mainly from small, overwintering water column populations while *Dolichospermum* spp. seems to originate from both akinetes and pelagic filaments (Wasmund et al., 2017; Suikkanen et al., 2010). The large improvement in seasonality when the lifecycle of cyanobacteria is modelled, as opposed to earlier modelling attempts that include only small winter populations, does

however, indicate that the separation into different lifecycle stages is of key importance for capturing the start and end of bloom.

Capturing the seasonality of cyanobacteria blooms is of great importance due to their impact on water quality as well as for obtaining better estimates of nitrogen fixation that contributes to eutrophication. This work constitutes a step forward for the modelling of cyanobacteria blooms in the Baltic Sea. The inclusion of CLC can with some further development be used to merge observations and modeling for obtaining better prognostic estimates of cyanobacteria blooms, which can be used for management purposes.

**Code availability**

The model code of the ocean model used for the simulations is publicly available from the Swedish Meteorological and Hydrological Institute, Norrköping, Sweden (https://www.smhi.se, E-mail: smhi@smhi.se).

**Data availability**

Model datasets displayed in the figures are publicly available: https://doi.org/10.5281/zenodo.5543392.

**Supplement link**

**Author contribution**

KE developed the RCO-SCOBI-CLC code and designed the experiments with the help of IH. KE also performed the model runs. MO provided the observational data on cyanobacterial biomass and calculated the estimates of nitrogen fixation based on previous *in situ* measurements. HEMM and EAR contributed to the design of the research. JH made the analysis and prepared the manuscript with input from all co-authors.

## Acknowledgments

The research presented in this study is part of the Baltic Earth program (Earth System Science for the Baltic Sea region, see http://www.baltic.earth) and was funded by the Swedish Research Council for Environment, Agricultural Sciences and Spatial Planning (FORMAS) within the project "Cyanobacteria life cycles and nitrogen fixation in historical reconstructions and future climate scenarios (1850-2100) of the Baltic Sea" (grant no. 214-2013-1449). Funding was also provided by the Swedish Research Council (VR) within the project "Reconstruction and projecting Baltic Sea climate variability 1850–2100" (Grant No. 2012–2117). The authors thank Ya-Wei Luo and two anonymous reviewers as well as the associate editor Yuan Shen for comments and suggestions that greatly improved the manuscript.

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

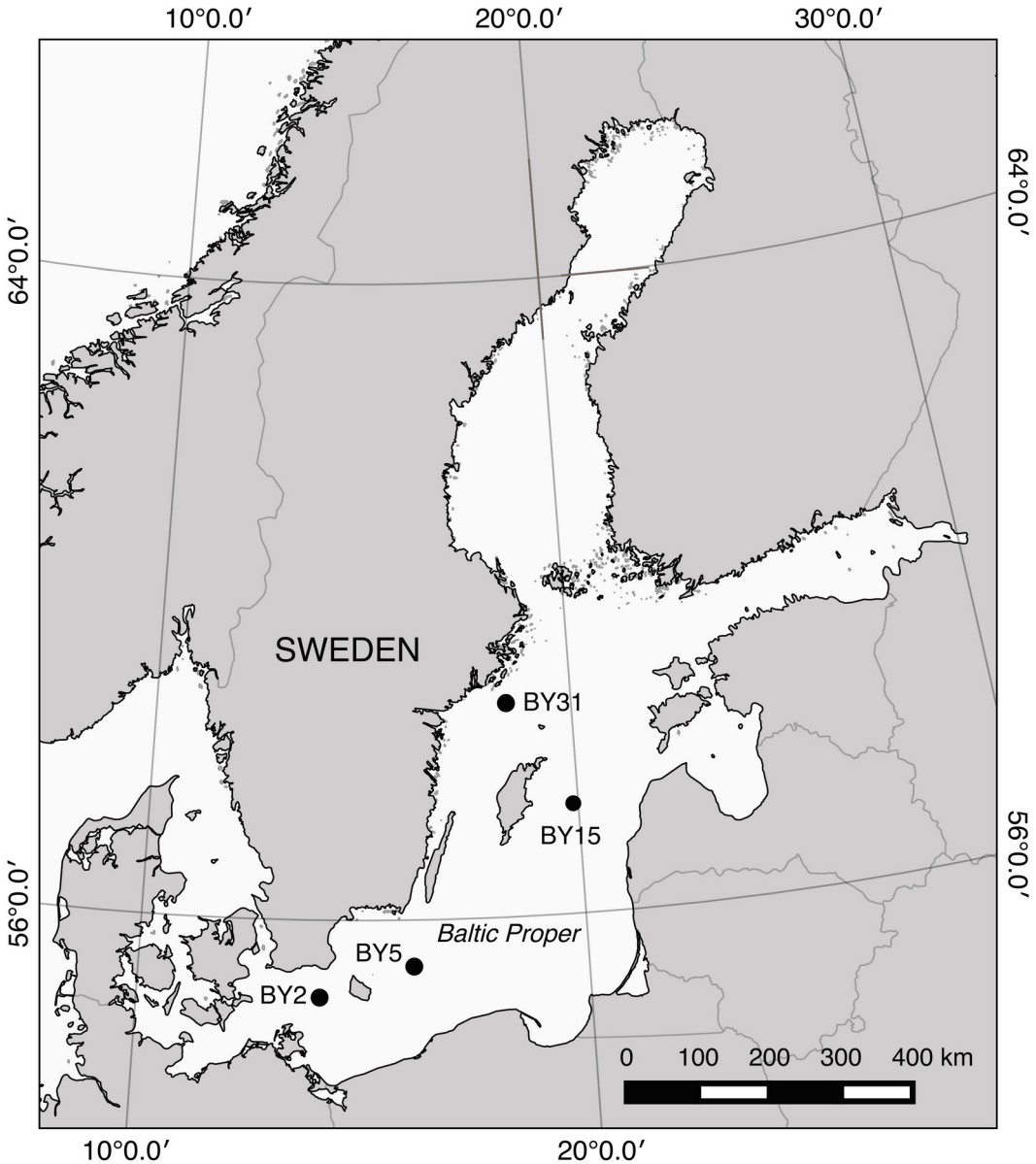

Figure 1. Map of the Baltic Sea. Baltic proper stations used in the study include BY31, BY15, BY5, and BY2.


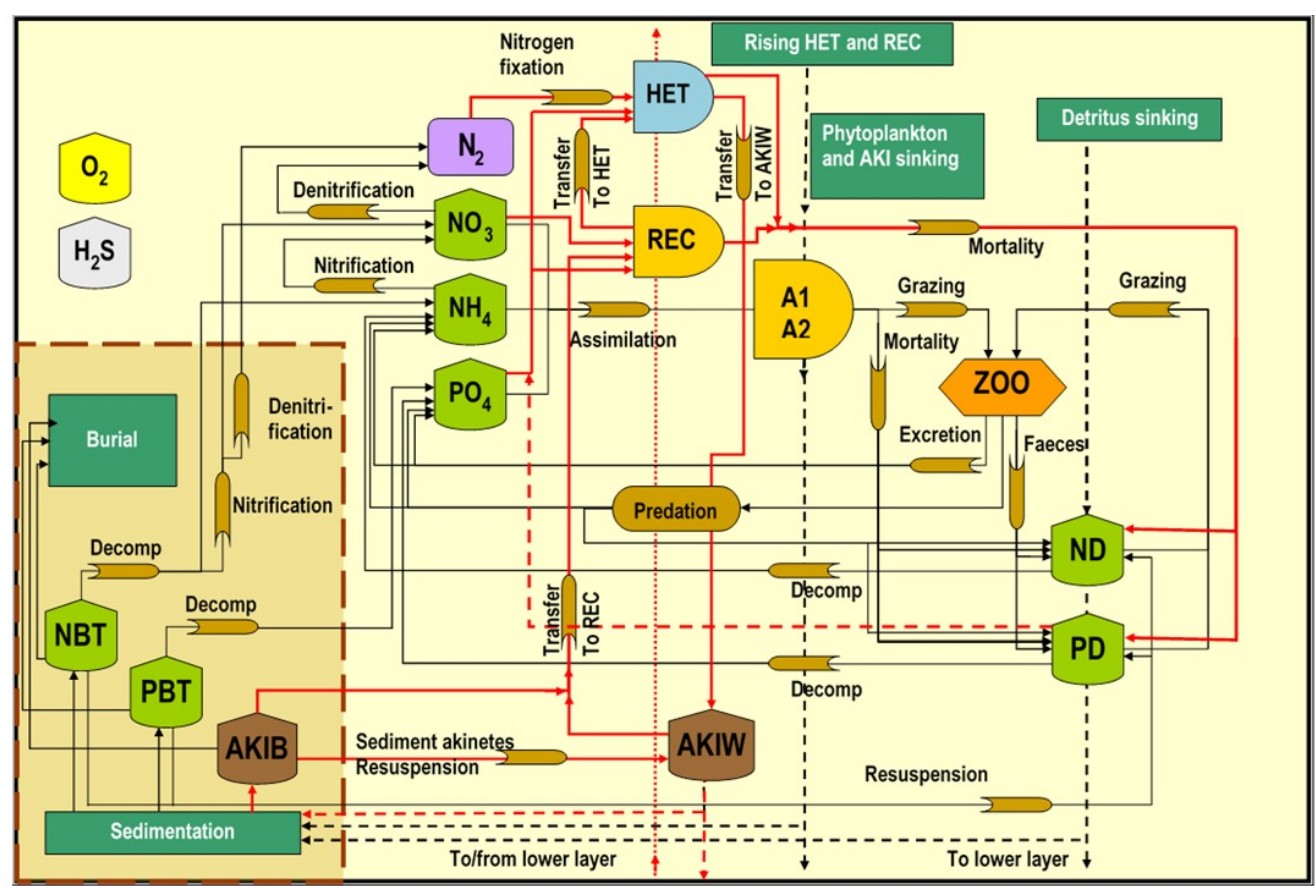

Figure 2. The SCOBI model (modified from Eilola et al., 2009) including the cyanobacteria life cycle model components indicated by red lines, vegetative cells with heterocysts (HET), Akinetes in water (AKIW) and in sediment ( AKIB), and Recruiting cells (REC). The inorganic nutrients nitrate, ammonia and phosphate are represented by NO3, NH4 and PO4, respectively. The phytoplankton groups A1 and A2 represent characteristics of diatoms and the flagellates and others. The bulk zooplankton ZOO grazes on phytoplankton A1 and A2 while the parameterized predation closes the system of equations. Nitrogen and phosphorus detritus are described by ND and PD, respectively. Oxygen dynamics are included and hydrogen sulfide concentrations are represented by negative oxygen equivalents (1 ml H2S l-1 = –2 ml O2 l-1). The process descriptions of oxygen and hydrogen sulfide are simplified for clarity. All abbreviations are described in Table 1.

| Category | Abbreviation | Long name | Description |
|---|---|---|---|
| **Autotrophs** | A1 | Autotroph 1 | A group of phytoplankton representing diatoms |
| | A2 | Autotroph 2 | A group of phytoplankton representing flagellates and others |
| **CLC** | REC | Recruiting cells | The growing but non-nitrogen-fixing stage of cyanobacteria |
| | HET | Heterocysts | The growing and nitrogen-fixing stage of cyanobacteria, vegetative cells with heterocysts |
| | AKIW | Akinetes water | The resting stage of cyanobacteria in water |
| | AKIB | Akinetes benthic | The resting stage of cyanobacteria in sediment |
| **Nutrients** | PO43- | Phosphate | Dissolved inorganic phosphorus in water |
| | NH4+ | Ammonium | Dissolved inorganic nitrogen in water |
| | NO3- | Nitrate | Dissolved inorganic nitrogen in water |
| | N2 | Nitrogen gas | Dissolved nitrogen gas in the water |
| **Detritus** | ND | Nitrogen detritus | Particulate organic nitrogen |
| | PD | Phosphorus detritus | Particulate organic phosphorus |
| **Benthic nutrients** | NBT | Benthic nitrogen | The nitrogen pool within the sediment |
| | PBT | Benthic phosphorus | The phosphorus pool within the sediment |
| | Burial | Burial | Burial of nutrients |
| **Zooplankton** | ZOO | Zooplankton | A group of zooplankton |
| **Oxygen** | O2 | Oxygen | Dissolved oxygen in water |
| | H2S | Hydrogen sulfide | Hydrogen sulfide is represented as negative oxygen |


Table 1. Description of abbreviations included in Figure 2.

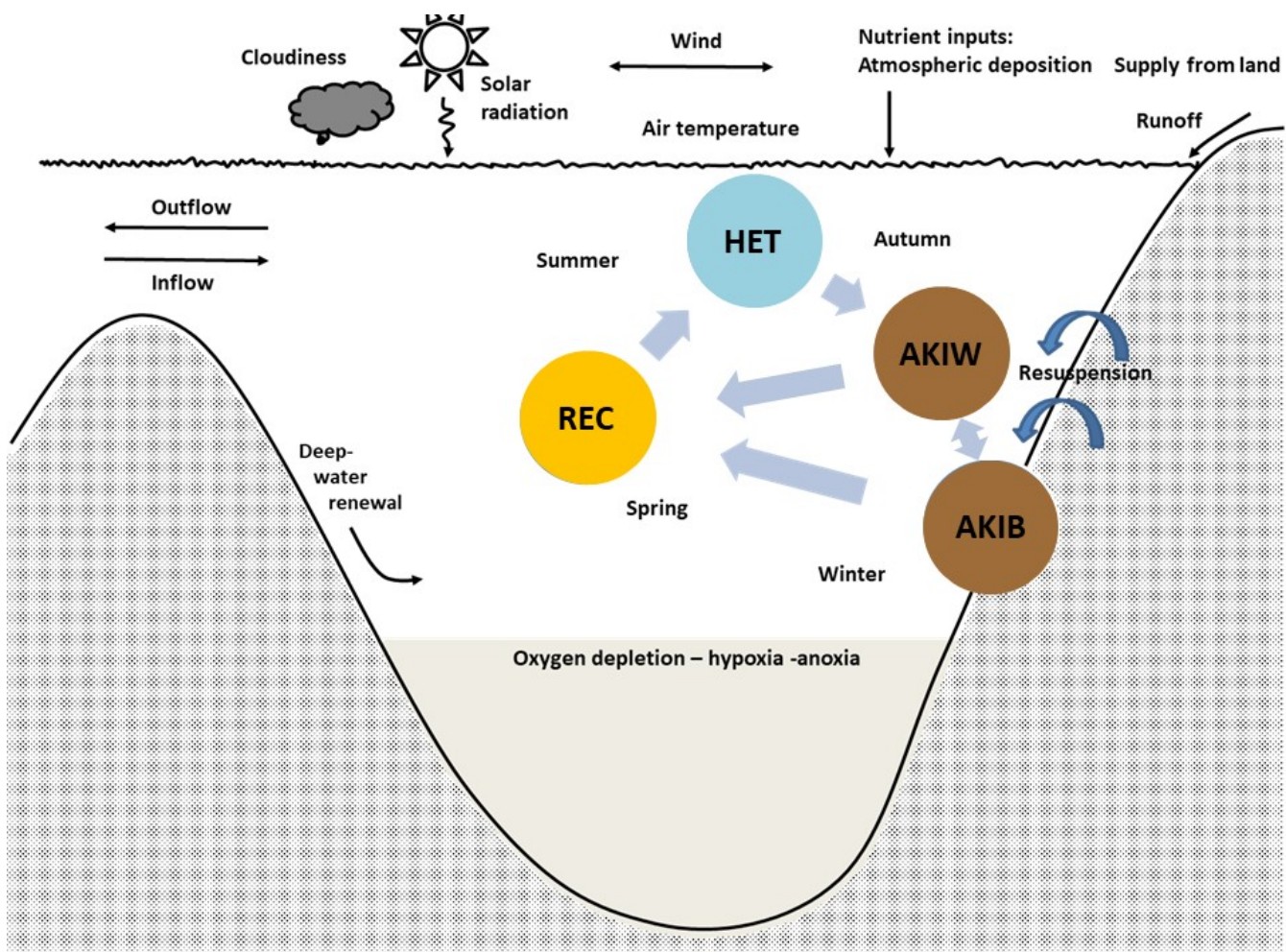

Figure 3. The simplified cyanobacteria life cycle used in the present model (modified after Hense and Beckmann, 2006; 2010) represents nitrogen-fixing filamentous, akinetes producing cyanobacteria with stage-dependent upward and downward velocity. The model includes three compartments, the nitrogen-fixing stage (HET), the resting stage of akinetes (AKI) and
the recruiting stage (REC). Occasions with resuspension may transfer akinetes from the sediment (AKIB) to the water (AKIW). Modified from Schneider et al. (2015) and Meier et al. (2019).

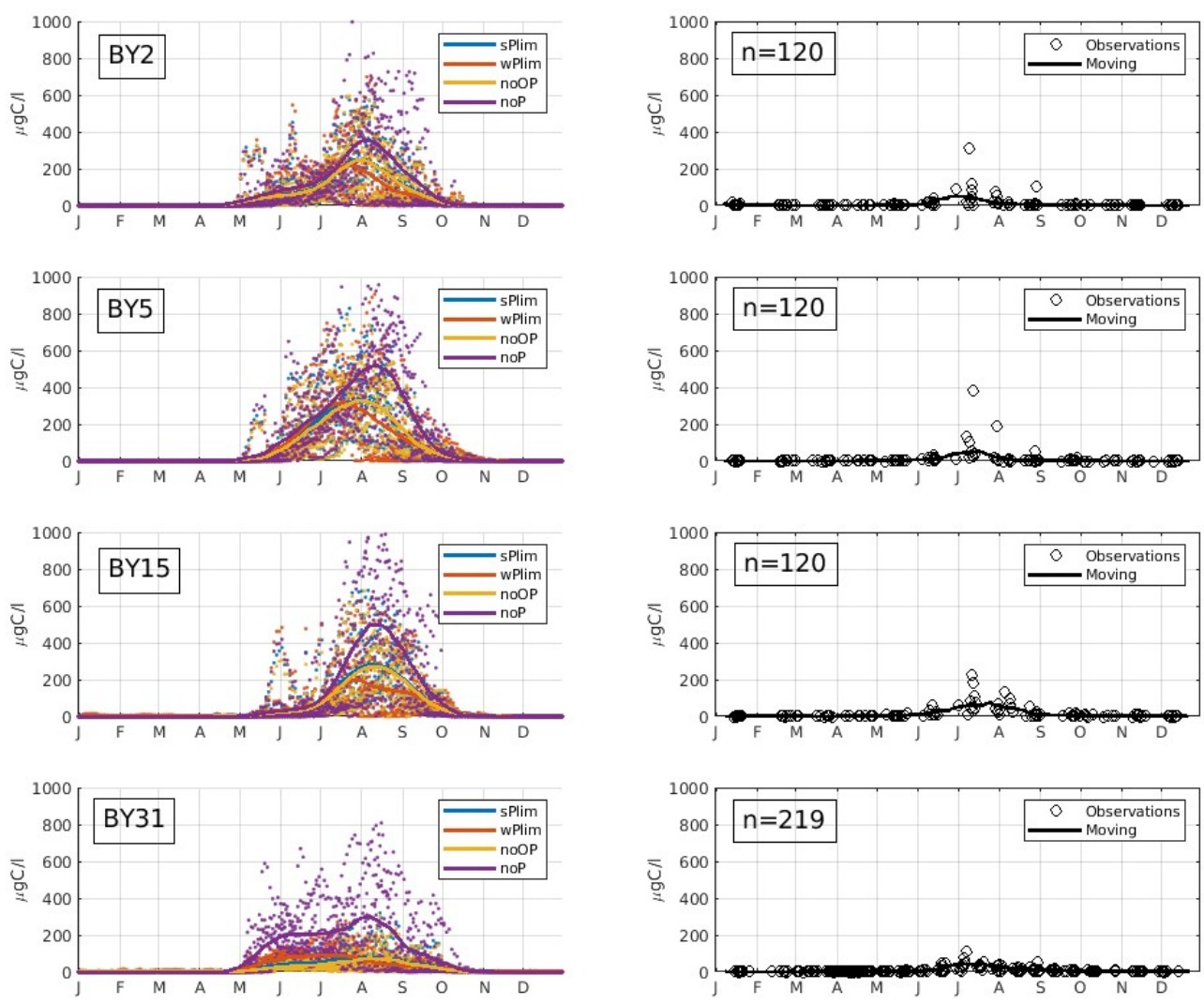


Figure 4. Mean seasonal cycle of cyanobacteria biomass (REC+HET) for four different stations in the Baltic proper (1999-2008). Left panels show SCOBI-CLC results and the right panels observations. The number of observations is indicated by "n" in the right hand panels. Dots show model output for every two days and solid lines represent the one month moving average.


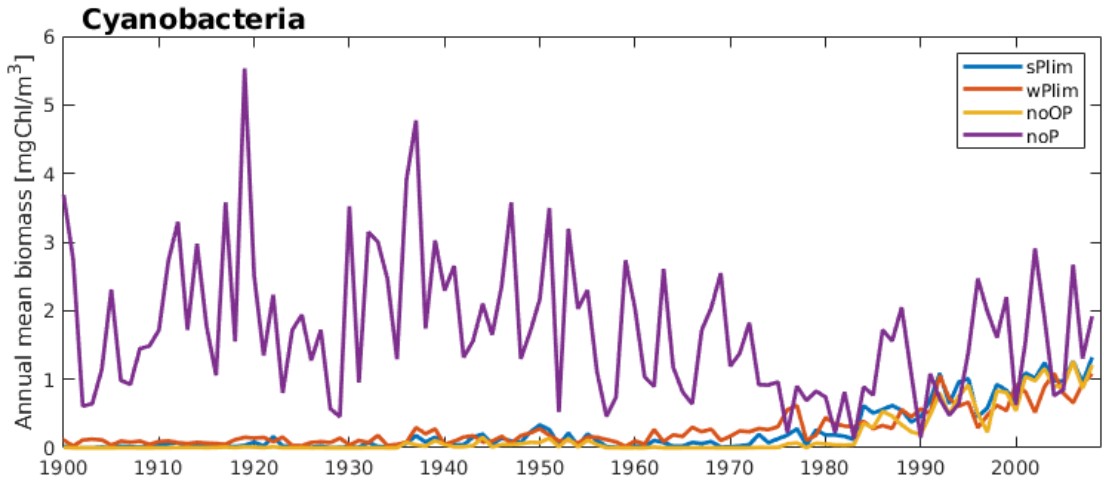

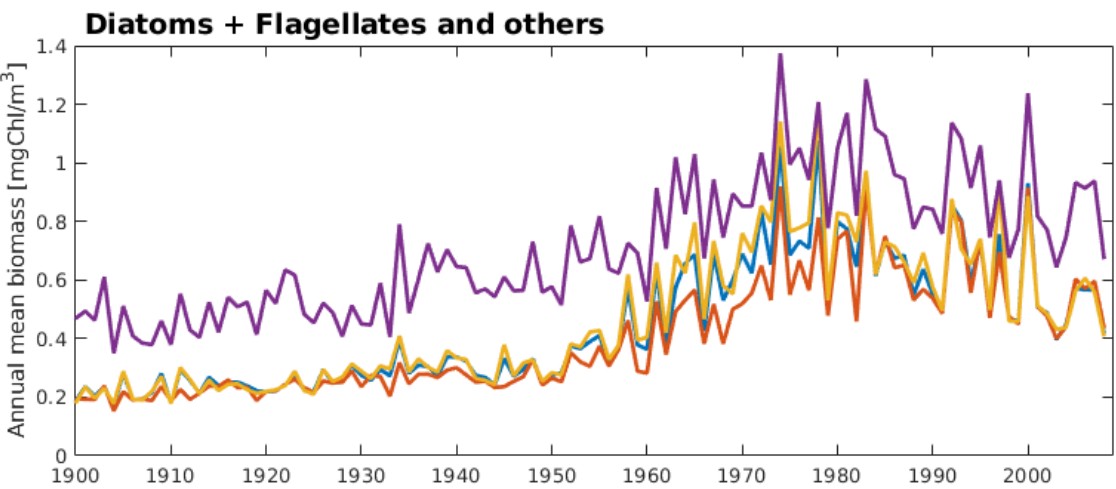

Figure 5. Simulated annual mean cyanobacteria biomass (REC+HET, upper) and the sum of annual mean biomass of functional types Diatoms and Flagellates and other autotrophic organisms (lower) at station BY15 for the four different phosphorus sensitivity experiments using SCOBI-CLC.

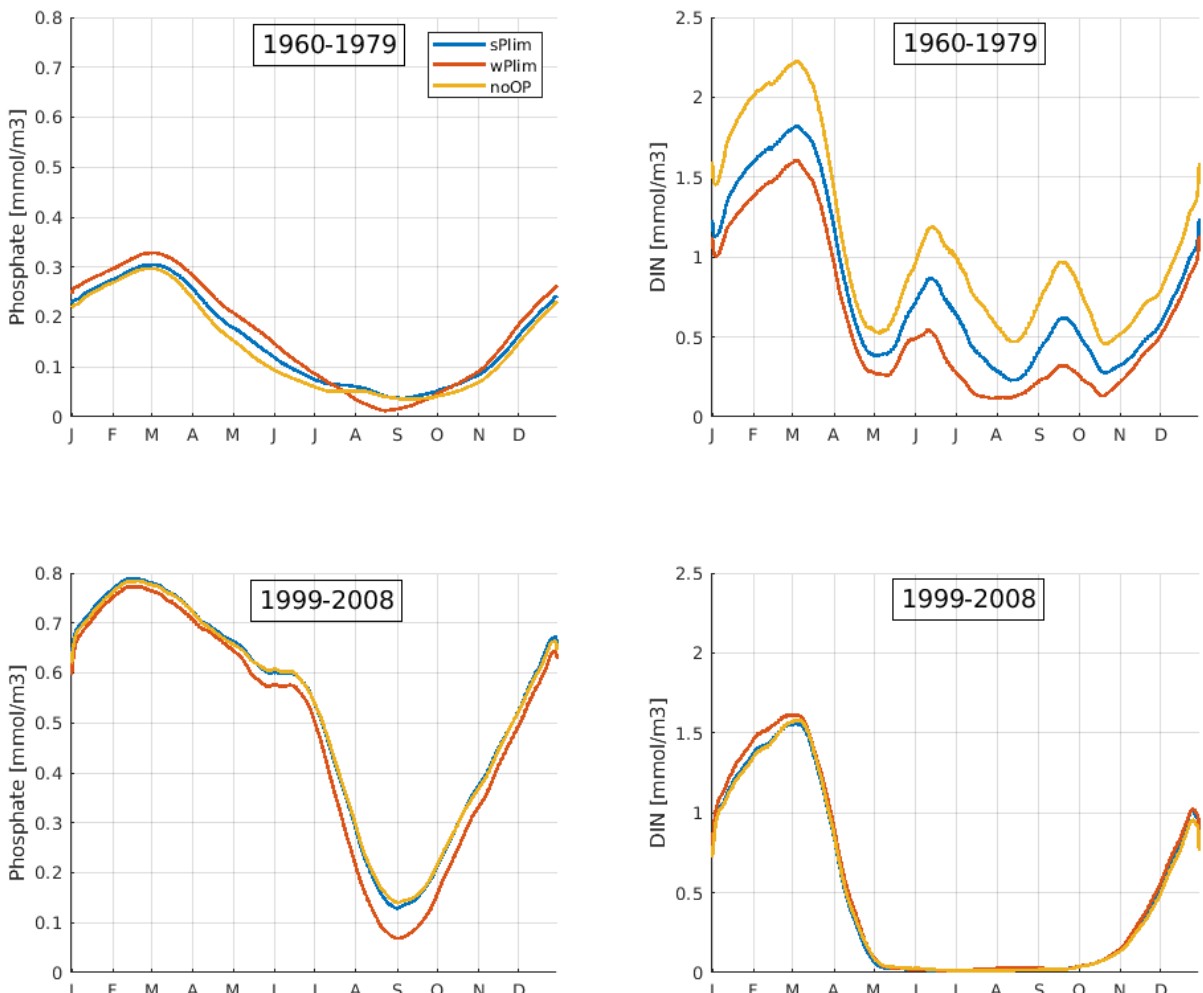

Figure 6. SCOBI-CLC results of the mean seasonal cycles of surface phosphate (left) and dissolved inorganic nitrogen; DIN (right) at station BY15 for the period 1960-1979 (upper) and 1999-2008 (lower). The data-points have been smoothed using a 1 month moving average.

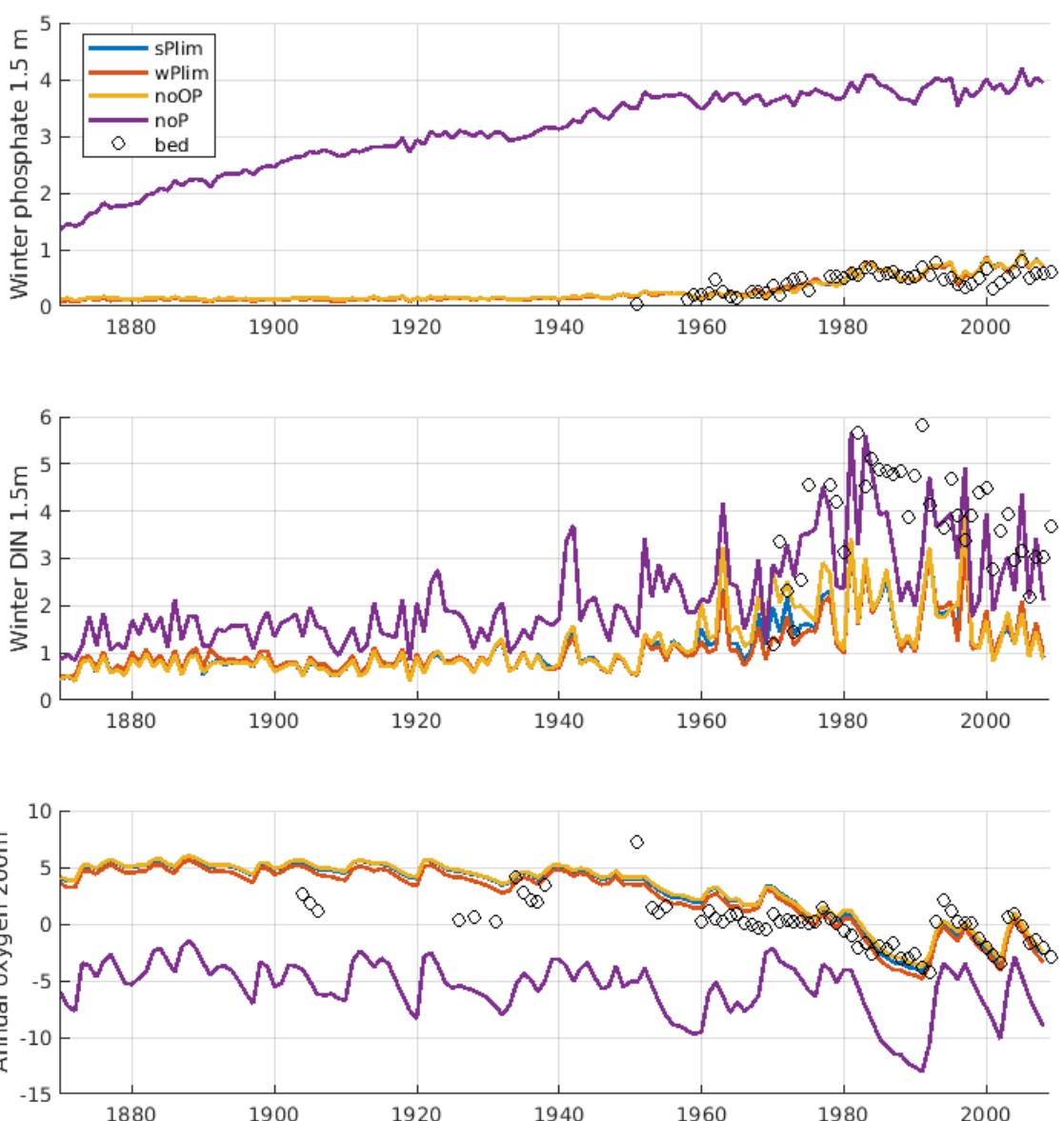

Figure 7. Evolution of winter (Jan-Feb) surface concentrations of phosphate (top), dissolved inorganic nitrogen (DIN; middle) and annual mean oxygen at 200m depth (bottom) at BY15. The solid lines show SCOBI-CLC results from the different phosphorus limitation experiments and the circles show observations from the Baltic Environmental Database (BED).

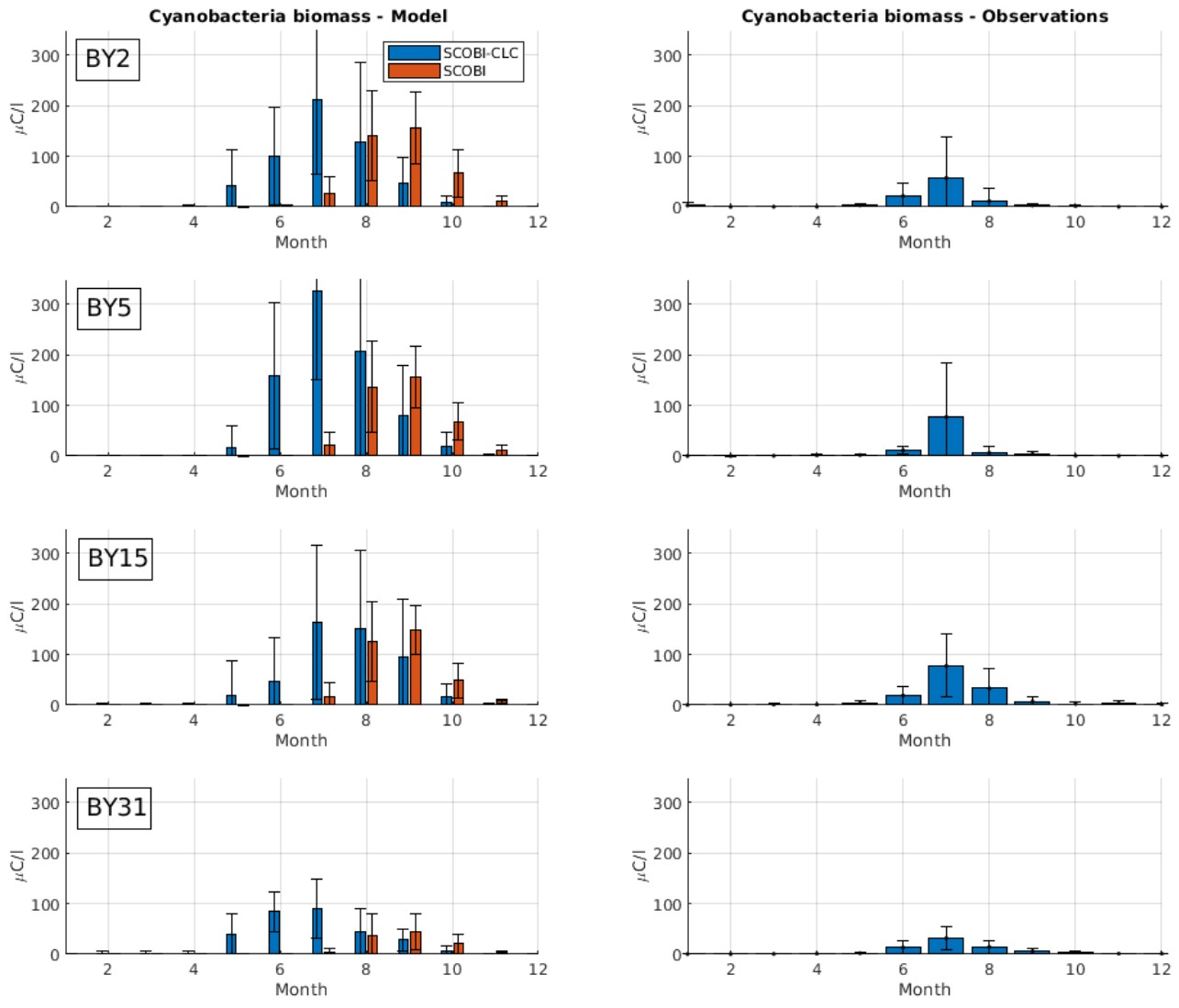

Figure 8. Simulated (left) and observed (right) monthly mean cyanobacteria biomass (REC+HET) for four Baltic proper monitoring stations over the years 1999-2008. Blue bars in the left column show SCOBI-CLC with the wPlim setting and the orange bars show SCOBI. Black lines show standard deviation.

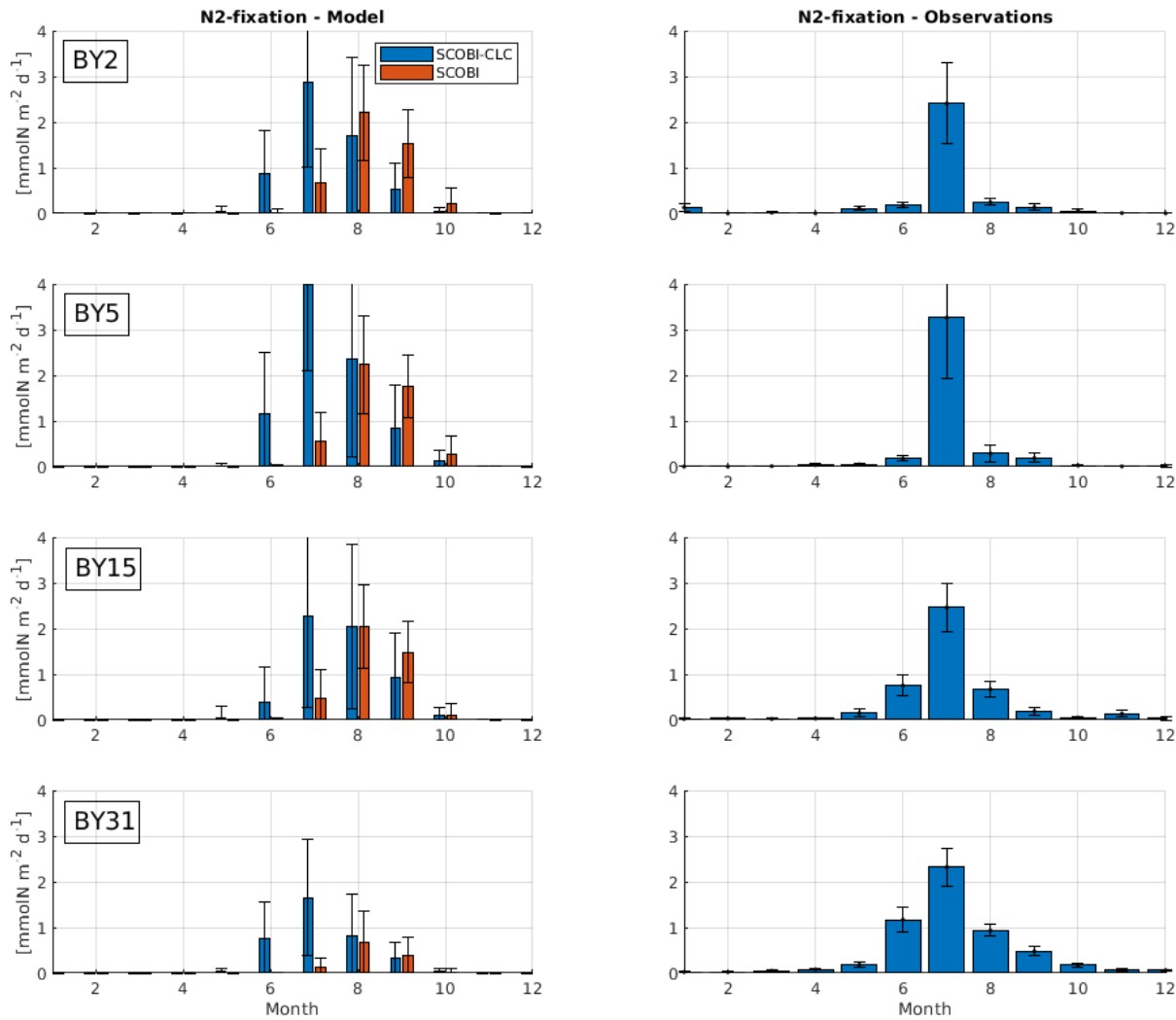

Figure 9. Model results (left) and observations (right) of monthly mean nitrogen fixation rates over the years 1999-2008 at different stations. Blue bars in the left column show SCOBI-CLC with the wPlim setting and the orange bars show SCOBI. Black lines show standard deviation.

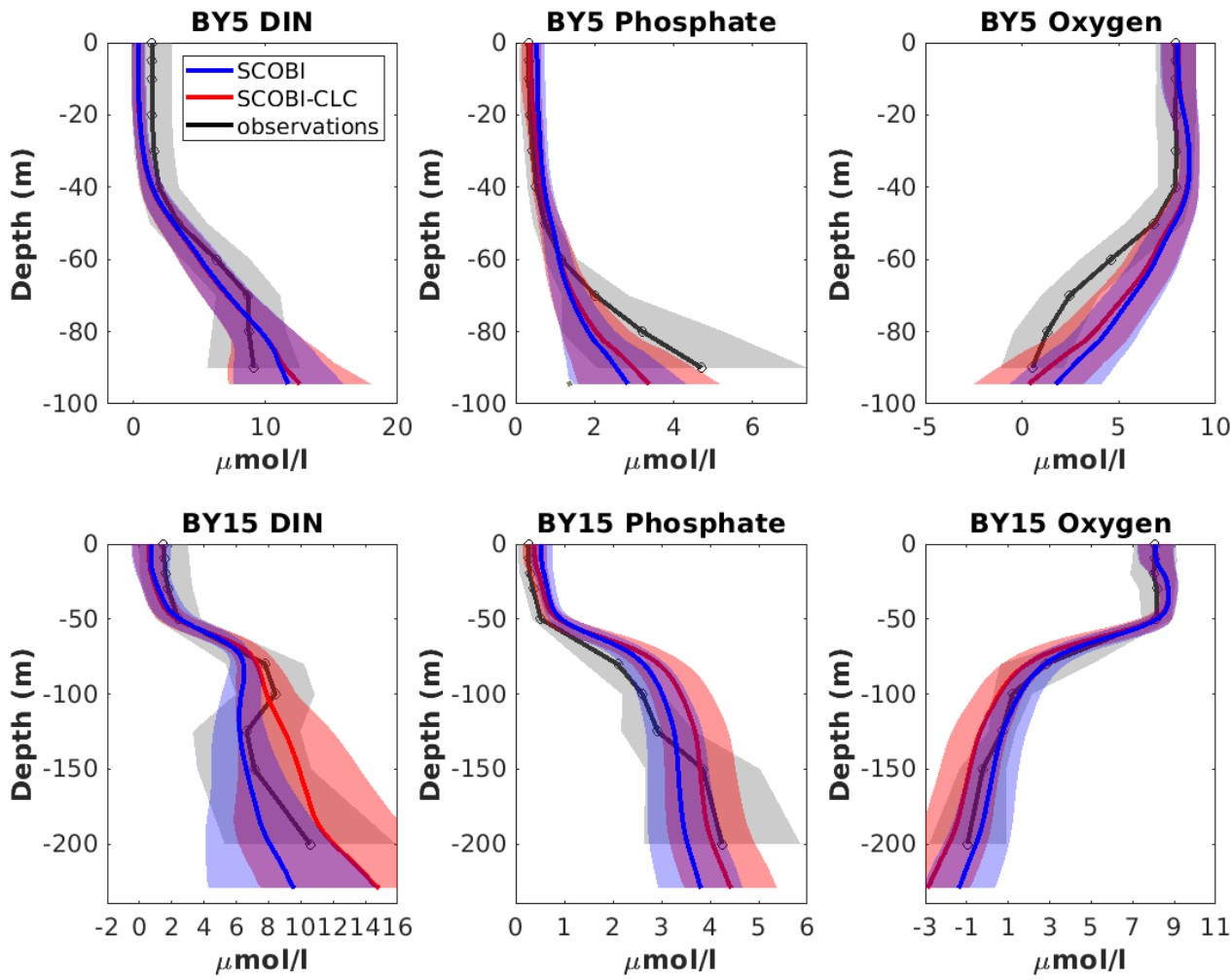

Figure 10. Model results with SCOBI (blue), with SCOBI-CLC model (red), and observations (black) of dissolved inorganic nitrogen (DIN), phosphate and oxygen at BY5 (upper panels) and BY15 (lower panels) averaged over the years 1976 to
2008. Observational data are from the SHARK database. Shaded areas represent standard deviation.