# Peer review of "Modeling cyanobacteria life cycle dynamics and historical nitrogen fixation in the Baltic proper"

_Biogeosciences, 2021_

## Author Comment (AC3)

[Figure]

*New Figure 2. The simplified cyanobacteria life cycle used in the present model (modified after Hense & Beckmann, 2006, 2010) representing $N_2$ fixing filamentous, akinetes producing cyanobacteria with stage-dependent upward and downward velocity. The model includes three compartments, the $N_2$-fixing stage (HET), the resting stage of akinetes (AKI) and the recruiting stage (REC). Modified from Schneider et al. (2015) and Meier et al. (2019).*

[Figure]

New Figure 3. The SCOBI model (modified from Eilola et al. 2009) including the cyanobacteria life cycle model components indicated by red lines, vegetative cells with heterocysts (HET), Akinetes in water (AKIW) and in sediment ( AKIB), and Recruiting cells (REC). The inorganic nutrients nitrate, ammonia and phosphate are represented by NO3, NH4 and PO4, respectively. The phytoplankton groups A1 and A2 represent characteristics of diatoms and the flagellates and others. The bulk zooplankton ZOO grazes on phytoplankton A1 and A2 while the parameterized predation closes the system of equations. Nitrogen and phosphorus detritus are described by ND and PD, respectively. Oxygen dynamics are included and hydrogen sulfide concentrations are represented by "negative oxygen equivalents (1ml H2S l-1 = –2 ml O2 l-1). The process descriptions of oxygen and hydrogen sulfide are simplified for clarity.

[Figure]

*Figure R1. Temperature limitation curve according to Eg. (8) in Table S.3. Temperature on the x-axis and degree of limitation on the y-axis. The highest value indicates the least limitation.*

---

## Author Response (AR1)

Replies to review comments

Reviewer 1:

The Hieronymus et al., have done through approach toward modeling phytoplankton and N2 fixation in Baltic Sea. The model seems to be well constructed developed upon the accumulated bodies of modeling and certainly provides new aspects of the regional modeling targeting this area. One challenge I had was to see how the cyanobacterial life cycle is simulated. A schematic and narrative would be useful. That said, I like how explicitly the filamentous bacteria is simulated, which is the unique feature of Baltic Sea. The manuscript is suitable Biogeosciences. The following are my comments hoping to improve the manuscript.

*Authors: Thank you for your thorough review and for pointing out that we need to be more clear in explaining the life cycle part of the model.*

*We have included a schematic figure of the CLC model (Fig. 3) and restructured and clarified the CLC description in section 2.3.*

Main text:

L20: in timing of -> in the timing of

*Authors: This has been corrected.*

L21: runs we -> runs, we

*Authors: This has been corrected*

L28: of its -> of their

*Authors: This has been corrected.*

L30: nitrogen fixing -> nitrogen-fixing (there are other cases below, which I would not mention)

*Authors: This has been corrected.*

L46: in bloom formation -> in the bloom formation

*Authors: This has been corrected*

L46: e.g. -> e.g.,   (there are other cases below, which I would not mention)

*Authors: This has been corrected*

L54: In the model growing -> in the model, growing (for clarity)

*Authors: Text removed*

L66: in abundance -> in the abundance

*Authors: Text removed.*

L73: gain understanding -> gain an understanding

*Authors: This has been corrected.*

L81: which has -> that has

*Authors: This has been corrected.*

L83: three dimensional -> three-dimensional

*Authors: This has been corrected.*

L88: the northern -> northern

*Authors: This has been corrected*

L101: tense should be consistent

*Authors: This has been corrected.*

L112: remove 'a' or make it to 'the'

*Authors: This has been corrected.*

L144: for the entire period -> for the entire period of (or put 1850 – 2008 in parenthesis)

*Authors: This has been corrected.*

L150: by very large burial -> by a very large burial (or 'the' very ...)

*Authors: This has been corrected.*

L175: For this work -> For this work,

*Authors: This has been corrected.*

L176: post processed -> post-processed

*Authors: This has been corrected.*

L186: includes also -> also includes

*Authors: This has been corrected.*

L189: true also -> also true

*Authors: This has been corrected.*

L193: A -> The (or remove 'A')

*Authors: This has been corrected.*

L197-210: There are different modeling experiment. I wonder which one is considered as default. Is there a model run that includes all the factors, which could be considered as default? I think it was done in the previous study? In any case, it would be useful to

compare these sensitivity analysis to be compared with the default, so I suggest putting the results from the default along with these simulations.

*Authors: There is no default in the original model. Since there has not been any previous 3D modeling efforts of the Baltic sea where the CLC model has been included, and the original model had unlimited P availability, we needed to evaluate P dependency before estimating nitrogen fixation and cyanobacterial biomass. Therefore, several runs using different settings were performed with the aim of producing the best fit to observations and in order to understand the CLC in the Baltic Sea and its effect, and depence on, the nutrient composition. From these sensitivity runs, one was chosen as the best fit (weak P limitation) and used in the estimates.*

*We have rewritten the introduction to include the lack of P dependency as a setting in the model and the importance of phosphorus for the cyanobacteria in the Baltic proper. Furthermore, we have restructured and partly rewritten Sections 2.6 and 3.1 so that this will be clearer.*

L220: which -> ,which

*Authors: This has been corrected.*

L225: in this case -> in this case,

*Authors: This has been corrected.*

L227: generating -> , generating

*Authors: This has been corrected.*

L228: faster growing -> faster-growing

*Authors: This has been corrected.*

L240: we -> , we (for clarity and improving readability)

*Authors: Text removed*

L245: I am having a hard time understanding what is meant by the life cycle model. Is that the diurnal cycle or is that longer cycle? A schematic (in addition to figure 2) and additional explanation (model summary with a few sentences) would be useful. I am suspecting  it is a seasonal cycle, so it would be nice if it is clearly defined here.

*Authors: The life cycle model is seasonal. This has been clarified in Section 2.3 as well in the new Fig. 3.*

L268: release -> the release

*Authors: This has been corrected.*

L301: we -> , we (for clarity and readability)

*Authors: This has been corrected.*

L301: seasonality of -> the seasonality of

*Authors: This has been corrected.*

L314: bloom forming -> bloom-forming

*Authors: Text removed.*

L320: however -> however,

*Authors: This has been corrected.*

Figure 2: Many different shapes are used. I wish to have a list of explanations for different shapes. Also, It is less clear where and how phytoplankton are represented. There seem to be multiple functional types of phytolankton but it is hard to see from the figure. I suggest another figure or panel to focus on phytoplankton functional types, as well as the life cycle of them since they seem to be key in the paper.

*Authors: We have revised the figure so that the colors of the CLC matches the colors of the CLC in the new Fig. 3. We have also included a table (Table 1) with abbreviations relating to Fig. 2. Since the specific shapes of the different variables in figure 2 is not crucial for the meaning of the flow chart and we would like to save the space in the manuscript, we choose not to explain the shapes. However, the specific colors  for the CLC variables are changed, to correspond to the colors of the figure 3 describing the CLC.*

Figure 3: Model seem to show much higher values than observations. I wonder what are the reasons.

*Authors: Figure 4 (old Figure 3) shows all different sensitivity runs and therefore also the runs with very high biomass values. However, the wPlim that we chose for the model results is not very far from the observations.*

*We have included a new figure (Figure 8) where the difference in monthly mean biomass is more visible. We have also included, in section 3.2, a discussion around the difference in biomass between model and observations.*

Figure 4: I am personally curious about how the population of N2 fixers change.

*Authors: We are a bit uncertain about this comment. Is it regarding how the community composition changes? Because the current figure, the upper panel, demonstrates how the whole population changes over time. Since the model only includes one group of cyanobacteria we can unfortunately not demonstrate how each taxa changes over time.*

Figure 5: The rate of nitrogen fixation seems to match despite the difference in biomass shown in Figure 3. I wonder what explains this. Also, I wish to see discussion on how Heterotrophic N2 fixation may alter the result.

*Authors: The strong coherence between model results and observed nitrogen fixation is somewhat surprising given the larger cyanobacteria biomass displayed by all model experiments compared to observations (new Fig. 4).*

*There are several potential causes for the deviation in carbon biomass estimations from the model and observations. The cyanobacteria biovolume from observations was used with different presumptions to estimate the nitrogen fixation rates and to calculate carbon concentrations, respectively. The modelled nitrogen fixation is calculated during the run from the growth of HET in the CLC model while the carbon content of cyanobacteria is calculated by the Redfield ratio between nitrogen and carbon in HET with a minor contribution also from REC. Hence, there are uncertainties in the calculations of carbon biomass from both observations and from model results.*

*It is not easy to change the Redfield C:N:P ratio that is used in the model since the results from the entire biogeochemical cycle including the oxygen consumption in the model is dependent on this ratio. There are other biogeochemical models with variable C:N:P ratios that might be used to analyze the impact from these processes further. Uncertainties in the comparison of models and observations stem also from the fact that observations are done on small water samples from an area that is covered by an*

*average value from a 3.7 km x 3.7 km grid in the model. We have included this discussion in Section 3.2.*

*Heterotrophic N2 fixation is extremely low in the Baltic Sea (e.g., Farnelid et al. 2013) and is therefore not included here. It has been demonstrated in the Baltic Sea that its three taxa that dominate the N2 fixation (Klawonn et al. 2016). Heterotrophic N2 fixation is neither included in the observations nor the model, and would probably not make any notable difference since it is so small.*

*We have included a discussion around this in section 3.2.*

Figure 6: How do these compare to the model simulation?

*Authors: These are from model simulations. This has been clarified.*

Figure 8: Could this be compared with observation?

*Authors: The data in red is from the model simulation wPlim and in black from observations. This has been clarified in the revised version of the manuscript both in the text and the figure legend.*

Supplementary material:

I wish to get the explanations behind (8). Why is it power of 4? Is that based on some previous studies?

*Authors: The equations for light and temperature limitation are adapted to Nemo-SCOBI from the original model by Beckmann and Hense (2004) and Hense and Beckmann (2006).*

I wish to get the reasoning behind (9). What is it formed with the addition of square termed in square root instead of the simple additions? Is that based on some previous studies?

*Authors: See previous answer.*

I wish to get some explanations behind equation (10) and (11), especially the reasoning of the mathematical formulas and qualitative interpretation of them.

*Authors: Eq. (10) describes the transition from the recruiting and vegetative state (REC) to the diazotrophic state (HET). The maximum growth rate ($s^{-1}$) of REC is larger than that*

*of HET but the growth (mmol m$^{-3}$ s$^{-1}$) is, in the previous state, also dependent on nitrogen. When the growth of HET is larger than that of REC a transition to HET occurs.*

*Eq. (11) describes the transition of HET to pelagic akinetes (AKIW). If the growth of HET is below a critical value, a transition to AKIW occurs.*

*We have deepened the model description and included a schematic of the CLC model (Fig. 3).*

Other points for discussion:

There are studies suggesting that heterotrophic bacteria may contribute to N2 fixation. I suggest considering discussing their effect on the overall N2 fixation in the Baltic Sea. The following papers may be useful: (Bentzon-tilia et al., 2015; Farnelid et al., 2013; Bentzon-Tilia et al., 2014; Chakraborty et al.; Pedersen et al., 2018).

*Authors: We agree that heterotrophic N2 fixation should be mentioned and a discussion around this has been included in section 3.2.*

N2 fixers (or nitrogenase) are known to be sensitive to O2. However, heterocysts have glycolipid layer which may protect them from O2. I think the hidden assumption in the model is that O2 does not matter to heterocysts. To support the assumption, the authors may consider citing (Inomura et al., 2017), as it shows that respiratory protection is not required for heterocysts; otherwise the rate of N2 fixation would be O2 dependent.

*Authors: We agree that heterocysts are designed to protect them from O2 and therefore is this model assumption correct. The suggested paper includes a totally different species (soil bacterium) so if needed we prefer to cite something closer to our study organisms. However, since the filamentous cyanobacteria in our study are photosynthetic they produce their own oxygen, and therefore always have O2 present around their cells, and why changes in O2 concentration does not affect the nitrogen fixation rates.*

References:

Bentzon-Tilia M, Farnelid H, Jürgens K, Riemann L. (2014). Cultivation and isolation of N2-fixing bacteria from suboxic waters in the Baltic Sea. FEMS Microbiology Ecology 88: 358–371.

Bentzon-tilia M, Severin I, Hansen LH, Riemann L. (2015). Bacteria Isolated from Estuarine Surface Water. Mbio 6: 1–11.

Chakraborty S, Andersen K, Visser A, Inomura K, Follows MJ, Riemann L. Quantifying nitrogen fixation by heterotrophic bacteria in sinking marine particles. Nature Communications Accepted.

Farnelid H, Bentzon-Tilia M, Andersson AF, Bertilsson S, Jost G, Labrenz M, et al. (2013). Active nitrogen-fixing heterotrophic bacteria at and below the chemocline of the central Baltic Sea. The ISME Journal 7: 1413–1423.

Inomura K, Bragg J, Follows MJ. (2017). A quantitative analysis of the direct and indirect costs of nitrogen fixation: a model based on Azotobacter vinelandii. The ISME Journal 11: 166–175.

Pedersen JN, Bombar D, Paerl RW, Riemann L. (2018). Diazotrophs and N2-Fixation associated with particles in coastal estuarine waters. Frontiers in Microbiology 9: 1–11.

*Author reply references:*

*Beckmann, A., & Hense, I. (2004). Torn between extremes: The ups and downs of phytopiankton. Ocean Dynamics, 54(6), 581–592. https://doi.org/10.1007/s10236-004-0103-x*

*Hense, I., & Beckmann, A. (2006). Towards a model of cyanobacteria life cycle-effects of growing and resting stages on bloom formation of N2-fixing species. Ecological Modelling, 195(3–4), 205–218. https://doi.org/10.1016/j.ecolmodel.2005.11.018*

*Klawonn, I., Nahar, N., Walve, J., Andersson, B., Olofsson M., Svedén, J.B., Littmann, S., Whitehouse, M.J., et al. 2016. Cell-specific nitrogen- and carbon-fixation of cyanobacteria in a temperate marine system (Baltic Sea). Environmental Microbiology 18: 4596–4609.*

**Reviewer 2:**

Major comments:

The paper conducts a 3-D modeling of Baltic Proper to simulate filamentous diazotrophic cyanobacteria and associated N2 fixation. The two major points that the authors appear to emphasize is that, the incorporation of cyanobacteria life cycle (CLC) dynamics and phosphorus dependence in the model greatly improve model performance in correctly simulating seasonality of the cyanobacteria biomass and N2 fixation. However, I found that the paper lacks clear focus, introduction, and thorough analyses. I was also puzzled by some results particularly from P dependence schemes.

*Authors: Thank you for this observation. We have revised the manuscript to become more focused and streamlined to the aims of the paper (to include CLC to a 3D model and perform phosphorus sensitivity runs to optimize specific to the Baltic Sea). Please see more detailed replies to the comments regarding the mentioned issues here below.*

(1) CLC appears to be one of the key issues the paper aims to resolve. Although the lead author and others have published a series of papers of CLC simulations, which making me not quite clear if CLC is still one of the key schemes that this paper would focus, at least both in the abstract (line 17-18) and conclusion (Line 301-304) the CLC is described as the main point of the paper. However, scientific background of life cycle of filamentous cyanobacteria is not sufficiently introduced in the paper, making it difficult to understand the different stages set in Method Section 2.3. Even in this section, CLC model part is not clearly described. There is Fig. 2 included in the paper which appears to be the structure of CLC and the model, but this figure is not referred and described. More importantly, model results of CLC (of different stages) are not shown. How the CLC improve the model is not analyzed and compared (such as to a model version without CLC or to the previous studies), except only two concluding sentences (line 244-245).

*Authors: It is true that CLC is the main focus of the paper since this is the first 3D modelling effort that includes the CLC in the Baltic Sea. Although the CLC model has been published before by two of the co-authors (Hense and Beckman 2006; 2010, Hense et al. 2013) this was neither combined with a 3D model nor specifically validated to observations from the Baltic proper. In these previous publications there was free P availability, which is not true for the Baltic proper, where P is very limiting for the filamentous cyanobacteria during the summer blooms together with weather conditions (e.g., Klawonn et al. 2016; Olofsson et al. 2016; Degerholm et al. 2006).*

*This has been included in the revised version of the manuscript, starting with the background of cyanobacteria life cycle stages and P limitation in the introduction.*

*We have clarified both that this is the first time CLC and 3D model is used as a combination in the Baltic proper and why we need to evaluate P dependency before estimating biomass and nitrogen fixation.*

*We have also added a comparison with a model run that excludes the CLC in section 3.2.*

(2) P dependence and weak P limitation (wPlim). I have problem to follow the key scheme (wPlim) that chosen as the main scheme. Half saturation concentration is very low ($10^{-6}$ nM), how it can be effective? Naturally there should be no complete absence of phosphate. The current half saturation of $10^{-6}$ nM is extremely low, several order of

magnitude lower below the detection limit. Indeed, Fig. 6. shows the lowest phosphate concentration is still much higher than 10^-6. How it can make substantial different results from noP as shown in such as Figs. 3 and 4? Even in these two Figures, Fig.3 and 4, wPlim results are not consistent (compared to other setups). For example, in Fig. 3 seasonal experiments, wPlim produce lower biomass for three of four sites (BY2\5\15). How in the Fig. 4 interannual variation, wPlim gives higher biomass?

*Authors: The difference between noP and wPlim is that noP requires no P at all to bloom while wPlim blooms as long as P is present, even in tiny (~10^-6) concentrations. This means that when all P is consumed, the cyanobacteria can no longer grow.*

*In the noP case, the end of bloom is completely dependent on the temperature and light availability (cf. Eq. (1), (2) and (6) in Table S3 and Table S5). Furthermore, there is no uptake or release of phosphorus by the cyanobacteria in this case which means that they do not affect the nutrient composition of the water column.*

*The relationship between the biomass and the P limitation scheme is not straight forward. It is not possible to say that wPlim will always generate the highest biomass compared to the other setups. The choice of P limitation not only affects the biomass but also the nutrient composition of ambient water which in turn affects the biomass of all functional types. Fig. 6 shows that during the early period, the phosphate concentrations were higher and the DIN lower compared to the other experiments generating higher cyanobacteria biomass. During the later period, DIN is completely depleted after the spring bloom in all experiments, while the phosphate is lowest in wPlim generating the lowest cyanobacteria biomass. Note that Fig 6 shows the mean seasonal cycle of phosphate over two different time periods at monitoring station BY15 only and says nothing about the overall minimum concentration.*

*We have deepened this discussion in Section 3.1 and restructured Section 2.6 so that the difference between the different experiments is easier to follow.*

Overall, the authors emphasize both CLC and wPlim schemes give better bloom timing (or seasonal pattern). However, even the "timing" of bloom is not quantitatively defined, and therefore the comparison and the conclusion they give better bloom timing is not supportive. For example, in Fig. 3, I cannot directly identify the difference of the start and end time of the bloom in each experiment. The authors should quantitatively define and show in numbers the start and termination of the blooms both in observations and the four experiments.

*Authors: To compare the results of the CLC model to our previous model version we have added a model run that excludes the CLC. We have added the results in the new Figures 8 and 9 which displays the monthly mean biomass and nitrogen fixation in the new and old model, and in observations. The difference in seasonal timing as well as the difference in biomass and nitrogen fixation is now clearer.*

Specific comments:

Section 2.2. Model structure Fig 2 should be referred and basic structure of the model sufficiently described.

*Authors: We have modified the SCOBI schematic and included a table that explains the abbreviations. We have also included a schematic that shows the CLC model. Section 2.3 that describes the CLC model has been restructured and rewritten to be more accessible and clear.*

Section 2.3. The text of this section should be reorganized to logically describe the stages and transitions.

*Authors: see previous answer.*

Line 117-119: Two versions? Or is it the "simplified version" of the "modified Version"?diazotro

*Authors: It is a mix of the original model of Hense and Beckmann (2006) and the simplified model of Hense and Beckmann (2010). We have clarified the model description in Section 2.6.*

Line 125-126, logically it is not justified why the differences among the species cannot influence the main patterns?

*Authors: Since we are using an average salinity and temperature preference range we can not look at specific regions where for example one of the taxa dominates because they might be outside the mean range. We will clarify this in the revised version of the manuscript. It might for example be difficult to apply our settings in a low salinity region of the Baltic Sea where we may have cyanobacteria that can grow in salinities down to 0, while the model has the lower range set to salinity 3.*

Line 132, the difference between AKIW and AKIB is not described.

*Authors: Akinetes are pelagic (AKIW) or benthic (AKIB) and can be transferred between these reservoirs through sinking and resuspension. We have clarified this in Section 2.3 but have also added a table which explains the abbreviations (Table 1).*

Line 139-140: "For the transition between AKI (AKIB and AKIW) and REC we prescribe a fixed germination window - from April 20 to the end of April": It is unclear. How the germination window defined? So, there is only one full life cycle each year? Before April 20, it is HET to AKI; and after end of April, it is always REC?

*Authors: The germination is defined in Eq. (30) in Table S.3. Between April 20 and the end of April, germination occurs at a constant rate times the AKI concentration. The transition from HET to AKIW is defined in Eq. (11) in Table S.3. It is dependent on the temperature and occurs when the growth of HET has fallen below a critical value. There is thus only one full cycle each year. We have clarified this in Section 2.3.*

Line 146, "Growth of HET and REC are inhibited under anoxic conditions." Why? Normally diazotrophs prefer anoxic conditions, right?

*Authors: We do not understand why they would prefer anoxic conditions. As they spend their active life stage in surface waters where there is enough light for them to perform photosynthesis there is plenty of oxygen around all the time, as they also produce oxygen themselves. Heterotrophic diazotrophs might prefer other conditions but the organisms in this paper are all photoautotrophs.*

Line 147, What is difference between this salinity dependent window and the above-mentioned time window (April 20-end)?

*Authors: The salinity dependence has little to do with the seasonality but is an effect of the observation that the optimum growth conditions of cyanobacteria occur in salinities between approximately 3 and 10 PSU. This span is taken to approximately represent the optimum span of N. spumigena, Aphanizomenon sp. and Dolichospermum spp. (Rakko and Seppälä, 2014). Its effect is more to limit the growth spatially than seasonally. We have clarified this in Section 2.3.*

Line 150, The range already described two sentences above.

*Authors: This has been corrected.*

Line 151-153, Between 11C and 28C, it increases linearly from 10% to 100%?

*Authors: The temperature limitation is defined by Eq. (8) in Table S.3 and is shown in the enclosed Figure R1. It is an adaptation to Nemo-SCOBI from the original model by Beckmann and Hense (2004).*

[Figure]

*Figure R1. Temperature limitation curve according to Eg. (8) in Table S.3. Temperature on the x-axis and degree of limitation on the y-axis. The highest value indicates the least limitation.*

Line 181-182, Nitrogen fixation rates appear to be an important observation parameter. How exactly calculated (estimated) from biomass in this paper?

*Authors: Detailed calculations can be found in Olofsson et al. (2021) as we also refer to in the method section, but we will also add a few more explaining sentences on the calculations in the revised version of the manuscript. Biovolume (mm3 L-1) of the three different species were obtained from the SMHI database and mean values of volume-specific measurements of nitrogen fixation were obtained from in situ measurements of thousands of cells of each of the three taxa across two summer seasons (From Klawonn et al. 2016, as referred to in the manuscript). Observed taxa-specific biovolume (mm3 L-1) were multiplied with the taxa-specific nitrogen fixation measurements per day (umol N mm3-1 d-1) to obtain nitrogen fixation rates per volume water per day (mmol N L-1 d-1), and further depth-integrated over 0-10 m (mmol N m-2 d-1) to obtain area-specific nitrogen fixation rates. These rates could then be summarized for the whole year and multiplied with the size of the Baltic Prover (200 000 km2) to provide nitrogen loads via nitrogen fixation by filamentous cyanobacteria (kton N yr-1).*

Fig. 3. Unclear the seasonal cycle is for earlier period (1960-1979), later period (1999-2008) or full period (1850-2010)?

*Authors: It is for the period 1999-2008. This has been clarified in the caption.*

Line 195: Diazotrophs tend to have much higher C:P or N:P ratio than normal phytoplankton.

*Authors: It is true that they have a flexible ratio which can be both above and below Redfield, but the difference is not huge. Ploug et al. 2010 and 2011 show a fixation ratio of 6.6 for Baltic Sea filamentous cyanobacteria for example, this is only slightly above Redfield of 6.5. There is a difference between diatoms and dinoflagellates as well (Menden-Deuer and Lessard 2000), but we can not include all differences in this model and have to make some simplifications.*

Line 252-253, How the N2 fixation is simulated? That may indicate wrong simulation of biomass-specific N2 fixation rate.

*Authors: The N2 fixation is a function of the temperature, light availability, N/P ratio and P. It is fully described in the appendix of Eilola et al. (2009). This has been clarified in Section 2.2.*

Also, the observed N2 fixation is derived from observed biomass (still unclear to me the method); is that unreliable?

*Authors: It is based on many measurements (thousands of cells across two seasons; From Klawonn et al. 2016) and observations over a long period of time (ca. 10 years of monitoring data from biweekly sampling in the Baltic Proper) so we would say it is fairly reliable. We will describe how it was estimated in more detail in the methods of the revised version of the paper (please see a more detailed reply to this comment above).*

Some format issue: such as some incorrect parentheses (line 47, 107, 148), missing unit (line 201), incorrect subscripts and superscripts (line 178, 182), inconsistent color codes of experiments across figures 3, 4, 6, 7. "Baltic proper" or "Baltic Proper"?

*Authors: This has been corrected.*

*Author reply references:*

*Beckmann, A., & Hense, I. (2004). Torn between extremes: The ups and downs of phytopiankton. Ocean Dynamics, 54(6), 581–592.*
*https://doi.org/10.1007/s10236-004-0103-x*

*Eilola, K., Meier, H. E. M., and Almroth, E. (2009). On the dynamics of oxygen, phosphorus and cyanobacteria in the baltic sea; a model study. Journal of Marine Systems 75, 163 – 184. doi:https://doi.org/10.1016/j.jmarsys.2008.08.009*

*Hense, I., & Beckmann, A. (2006). Towards a model of cyanobacteria life cycle-effects of growing and resting stages on bloom formation of N2-fixing species. Ecological Modelling, 195(3–4), 205–218. https://doi.org/10.1016/j.ecolmodel.2005.11.018*

*Hense, I. and Beckmann, A. (2010). The representation of cyanobacteria life cycle processes in aquatic ecosystem models. Ecological Modelling 221, 2330 – 2338. doi:https://doi.org/10.1016/j.ecolmodel.*

*Hense, I., Meier, H. E. M., & Sonntag, S. (2013). Projected climate change impact on Baltic Sea cyanobacteria: Climate change impact on cyanobacteria. Climatic Change, 119(2), 391–406. https://doi.org/10.1007/s10584-013-0702-y*

*Klawonn, I., Nahar, N., Walve, J., Andersson, B., Olofsson M., Svedén, J.B., Littmann, S., Whitehouse, M.J., et al. 2016. Cell-specific nitrogen- and carbon-fixation of cyanobacteria in a temperate marine system (Baltic Sea). Environmental Microbiology 18: 4596–4609.*

*Menden-Deuer, Susanne, Lessard, Evelyn J., (2000), Carbon to volume relationships for dinoflagellates, diatoms, and other protist plankton, Limnology and Oceanography, 3, doi: 10.4319/lo.2000.45.3.0569.*

*Olofsson, M., Klawonn, I., and Karlson, B. (2021). Nitrogen fixation estimates for the Baltic Sea indicate high rates for the previously overlooked Bothnian Sea. AMBIO 50(1): 203-214.*

*Ploug, H., Adam, B., Musat, N., Kalvelage, T., Lavik, G., Wolf-Gladrow, D., and Kuypers, M.M.M.( 2011). Carbon, nitrogen and O2 fluxes associated with the cyanobacterium Nodularia spumigena in the Baltic Sea. ISME Journal 5: 1549– 1558.*

*Ploug, H., Musat, N., Adam, B., Moraru, C.L., Lavik, G., Vagner, T., Bergman, B., and Kuypers, M.M.M. (2010). Carbon and nitrogen fluxes associated with the cyanobacterium Aphanizomenon sp. in the Baltic Sea. ISME Journal 4: 1215–1223.*

*Rakko, A. and Seppälä, J. (2014). Effect of salinity on the growth rate and nutrient stoichiometry of two Baltic Sea filamentous cyanobacterial species. Estonian Journal of Ecology doi:10.3176/eco.2014.2.01*

**Reviewer 3:**

Jenny Hieronymus et al. Modeling cyanobacteria life cycle dynamics and historical nitrogen fixation in the Baltic Sea

This study incorporated a cyanobacterial life cycle model with phosphorus dependency, which improved the prediction of diazotrophic cyanobacterial blooms in the Baltic Sea. The research is quite interesting and challenging; however, I found the whole manuscript

lacks a clear hypothesis, clear clarification of why phosphorus is important, and the interpretation of results is not deep enough. I could see that the authors were trying to explain the methodology as it is a complicated study, however I got lost easily as there is not a clear approach or conceptual diagram to lead the readers. I have also got a few major concerns as listed below.

*Authors: Thank you for this observation, we have clarified our aims of the paper better in the introduction. We have revised the introduction to introduce the phosphorus dependency on an earlier stage, and also included a conceptual image of the CLC (Figure 3).*

*To clarify, we have for the first time included the CLC model into a 3D model for the Baltic proper. Previously the CLC model has been used by itself (Hense and Beckman 2006; 2010) and together with a 1d water column model (Hense et al. 2013). The P dependency of cyanobacteria has not been previously included, and since phosphate is limiting nitrogen fixation in the Baltic proper during summer (Degerholm et al. 2006, Olofsson et al. 2016 etc.) the level of P dependency needed to be evaluated to not completely overestimate the biomass of cyanobacteria. As our study demonstrated in the model experiment noP (which is reflecting the settings in the previous studies), the biomass is far above observed levels and therefore discarded as a suitable setting for the Baltic proper. Instead, we found wPlim to be closest to observations in timing and magnitude of biomass and this was chosen for the estimates (as described in lines 240-241). We have clarified this in the introduction and in the results.*

Introduction

From the manuscript it is not clear to me phosphorus utilization of the diazotrophic species is important in the Baltic sea, and what critical roles P plays in the dominance of the three species.

*Authors: We have added an explanation on this in the introduction.*

Methods

Fig. 2 seems very complicated and busy to me, and I cannot tell what processes the authors have modelled and tried to test. What is your hypothesis? To someone who is not familiar with CLC model, I am suggesting the authors making Fig. 2 easier to follow, also by adding a conceptual diagram to illustrate what life cycle really means – what are the physiological processes, timescale, input conditions and output, etc.

*Authors: We have included a schematic of the CLC (Figure 3) in parallell parallel to Figure 2. We have also color coded the CLC parts of Figure 2 and added a table with abbreviations (Table 1).*

Please also specify why CLC model needs to be modified to include P utilization. Did you mean by superior P uptake, P storage or DOP scavenging startegies?

*Authors: Phosphorus dependency has not been considered in previous versions of the CLC model but must be considered in the Baltic proper where P is often limiting the growth of filamentous cyanobacteria (e.g., Klawonn et al. 2016, Olofsson et al. 2016; Degerholm et al. 2006). We have rewritten the introduction to better explain this focus.*

L100 – why some of the predicted temperature were much higher? Please kindly explain the reason behind it.

*Authors: The temperature is well represented by the model. Slightly higher temperatures can be found in the upper parts of the halocline at BY15 (central Baltic sea, Fig. 10 in Meier et al., 2018). The reason is not clear but an exact reconstruction of the past is not to be expected by any model. For further details, please refer to Meier et al. (2018).*

L110 – But you could already see cyanobacterial species vary in physiology from a great many publications. I wonder if it could be better to allocate a range or different C:N:P ratios for the modelled species, maybe a sensitivity analysis could help you find out whether this ratio really matters for the simulated outcomes.

*Authors: We will think of this for the future, but for now it is too complicated. We have included a discussion around different C:N:P ratios in section 3.2.*

L125 – I am sure the internal nutrient quotas also affect the growth and life cycle transitions; however, I cannot tell if you have included internal nutrient quotas impacts. A schematic including how processes involved in the model, alongside the methodology of this manuscript may help clarify the uncertainty here.

*Authors: Internal nutrient and energy quotas were included in the original CLC model by Hense and Beckmann (2006). In a following publication (Hense and Beckmann, 2010), they constructed a simplified model, where the internal quotas were excluded, with the aim of obtaining a model efficient enough to be included in a 3d climate model. The model that we have used is an adaptation of their simplified model but where they separated the diazotrophic and non-diazotrophic stages into a two compartment model, we have instead summed up the recruiting and the vegetative (growing but without heterocysts) stage (REC) and obtained a four compartment CLC model including pelagic akinetes (AKIW), benthic akinetes (AKIB), recruiting and vegetative non-diazotrophic cells (REC) and cells with heterocysts (HET). We will include a new graphic of the CLC (Figure 3).*

Results and discussion

L565 – There are some extreme biomass values that were not predicted, why is that?

*Authors: The colored dots in Fig. 3 show the two daily values of simulated biomass for the different experiments. The high frequency of the model output compared to the maximum sampling frequency of once every two weeks for the observations, generates a higher probability to capture extreme highs and lows. That being said, the simulated biomass is higher in the model compared to the observations even for the best fit simulation wPlim. There could be many reasons for that such as a too high or low C:N ratio, or non optimal choices of other constants. We have included a discussion around this in Section 3.2. The difference in biomass and nitrogen fixation between the model and observations is now also clearer in the new Figures 8 and 9 that show the monthly means.*

*Author reply references:*

*Degerholm, J., Gundersen, K., Bergman, B., & Söderbäck, E. (2006). Phosphorus-limited growth dynamics in two Baltic Sea cyanobacteria, Nodularia sp. and Aphanizomenon sp. FEMS Microbiology Ecology, 58(3), 323–332.*
*https://doi.org/10.1111/j.1574-6941.2006.00180.x*

*Klawonn, I., Nahar, N., Walve, J., Andersson, B., Olofsson M., Svedén, J.B., Littmann, S., Whitehouse, M.J., et al. 2016. Cell-specific nitrogen- and carbon-fixation of cyanobacteria in a temperate marine system (Baltic Sea). Environmental Microbiology 18: 4596–4609.*

*Meier, H. E. M., Eilola, K., Almroth-Rosell, E., Schimanke, S., Kniebusch, M., Höglund, A., Pemberton, P., Liu, Y., Väli, G., & Saraiva, S. (2018). Disentangling the impact of nutrient load and climate changes on Baltic Sea hypoxia and eutrophication since 1850. Climate Dynamics. https://doi.org/https://doi.org/10.1007/s00382-018-4296-y*

*Olofsson, M., Egardt, J., Singh, A., and Ploug, H., (2016). Inorganic phosphorus enrichments in Baltic Sea water has large effects on growth, carbon fixation, and N2 fixation by Nodularia spumigena. Aquatic Microbial Ecology 77: 111–123.*

**List of relevant changes**

- The introduction has been restructured and streamlined to the main aims of the paper: the introduction of the cyanobacteria life cycle in a 3d model of the Baltic proper, and the importance of phosphorus limitation.
- The methods section 2.3 has been rewritten to more clearly and logically explain the new CLC model.
- A new CLC schematic has been added (Figure 3) to provide a clearer view of the CLC and CLC seasonality.
- Figure 2 has been changed so that the colors of the CLC correspond to the colors in the new Figure 3.
- A table (Table 1) has been added to explain the abbreviations in Figure 2.
- Section 2.5 (Observations) has been expanded to include a better explanation of the nitrogen fixation calculations.
- Section 2.6 (Phosphorus dependence) has been restructured and rewritten to better explain the difference between the phosphorus sensitivity experiments.
- Section 3 (Results) has been rewritten and structured more logically so that it starts with the different phosphorus sensitivity experiments and ends with a comparison with observations and a model run with the old setup excluding the CLC.
- A new model run has been added using the old SCOBI setup that does not include the CLC. Results are shown in Figures 8, 9 and 10.
- A new figure (Figure 8) has been added that shows the monthly mean biomass. The figure makes the improvement in seasonality between a model with and without CLC clearer.

---

## Author Response (AR2)

**Reply to reviewer #1**

Regarding below:
"I wish to get the explanations behind (8). Why is it power of 4? Is that based on some previous studies?
Authors: The equations for light and temperature limitation are adapted to Nemo-SCOBI from the original model by Beckmann and Hense (2004) and Hense and Beckmann (2006)."
I took a look at both paper but I could not find the same equation. I suggest the authors cite specific equation in these papers. If the formation is modified, I suggest that the author show derivation in supplementary material. Also, the previous papers do not seem to clarify why it is power of 4, which still remains question in this paper. I suggest that the author state why the authors chose power of 4. If it is imply empirical, that is fine too, but I suggest that it is stated.

***Authors:*** *Thank you for your observation. The temperature dependence of the cyanobacteria growth rate was designed to represent the observational data presented in Lethimäki et al. (1997). It closely resembles that of Hense and Beckmann (2006) but was redesigned as the original formulation was found to be numerically unstable in the 3d framework. We have added a few lines clarifying this in the manuscript (lines 163 to 167). We have also added references in the Supplementary material.*

***Author References:***

Lehtimäki, J., Moisander, P., Sivonen, K., and Kononen, K.: Growth, nitrogen fixation, and nodularin production by two baltic sea cyanobacteria. Appl Environ Microbiol., 63(5), 1647-1656, doi:10.1128/aem.63.5.1647-1656, 1997.

Hense, I. and Beckmann, A.: Towards a model of cyanobacteria life cycle—effects of growing and resting stages on bloom formation of N2-fixing species, Ecol. Model., 195, 205 – 218, https://doi.org/10.1016/j.ecolmodel.2005.11.018, 2006.

**List of relevant changes:**

- An explanation to the origin of the temperature limitation equation has been added to Section 2.3.
- References to the temperature limitation equation (Table S.3 Eq. (8)) and the light limitation equation (Table S.3 Eq. (9) and (27)) have been added.
- A reference list has been added to the supplementary material.